# An assembly of nuclear bodies associates with the active VSG expression site in African trypanosomes

James Budzak[1], Robert Jones[1], Christian Tschudi [2], Nikolay G. Kolev[2] & Gloria Rudenko [1✉]

A Variant Surface Glycoprotein (VSG) coat protects bloodstream form *Trypanosoma brucei*. Prodigious amounts of VSG mRNA (~7-10% total) are generated from a single RNA polymerase I (Pol I) transcribed VSG expression site (ES), necessitating extremely high levels of localised splicing. We show that splicing is required for processive ES transcription, and describe novel ES-associated *T. brucei* nuclear bodies. In bloodstream form trypanosomes, the expression site body (ESB), spliced leader array body (SLAB), NUFIP body and Cajal bodies all frequently associate with the active ES. This assembly of nuclear bodies appears to facilitate the extraordinarily high levels of transcription and splicing at the active ES. In procyclic form trypanosomes, the NUFIP body and SLAB do not appear to interact with the Pol I transcribed procyclin locus. The congregation of a restricted number of nuclear bodies at a single active ES, provides an attractive mechanism for how monoallelic ES transcription is mediated.

[1] Department of Life Sciences, Sir Alexander Fleming Building, Imperial College London, South Kensington, London SW7 2AZ, UK. [2] Department of Epidemiology of Microbial Diseases, Yale School of Public Health, New Haven, CT 06536, USA. ✉email: gloria.rudenko@imperial.ac.uk

The African trypanosome *Trypanosoma brucei* is spread by tsetse flies to the mammalian bloodstream, thereby causing African Sleeping sickness and the livestock disease nagana[1]. A protective coat comprised of ~$10^7$ Variant Surface Glycoprotein (VSG) molecules is crucial for its survival as an extracellular pathogen[2]. *T. brucei* encodes >2500 *VSG* genes, of which only one is expressed at a time from one of ~15 *VSG* expression sites (ES)[3,4]. Compared with more widely studied eukaryotes, *T. brucei* has highly unusual molecular biology features, including limited transcriptional control. The genome predominantly consists of extensive arrays of polycistronic transcription units, which are constitutively transcribed by RNA polymerase II (Pol II). Primary transcripts are matured into mRNAs through coordinated *trans*-splicing of a capped spliced leader (SL) RNA exon and polyadenylation[5]. Remarkably, *T. brucei* is the only eukaryote known to transcribe protein-coding genes by Pol I, which normally exclusively transcribes rDNA. These Pol I-transcribed genes include ESs in bloodstream form (BF) trypanosomes, and procyclin genes expressed in the procyclic form (PF) trypanosomes present in the insect vector[6]. Highly stringent monoallelic exclusion ensures that only one ES is transcribed at a time, in an extra-nucleolar Pol I-enriched Expression site body (ESB)[7]. Trypanosomes forced to simultaneously express two ESs, dynamically share a single ESB[8].

VSG is an example of 'extreme biology'. VSG is produced from a single active *VSG* gene, yet comprises by far the most abundant mRNA and protein (~7–10% total) in BF *T. brucei*. These extraordinarily high expression levels are facilitated by high rates of transcription initiation by Pol I, whereby subsequent *trans*-splicing results in the generation of functional mRNA[9]. We investigated how these phenomenal levels of mRNA expression from a single *VSG* gene are achieved and explored the role of nuclear bodies including Cajal bodies in maintaining what must be extremely high levels of localised *trans*-splicing.

Cajal bodies (CB) are non-membrane-bound nuclear bodies playing an important role in assembling splicing components, including the small nuclear ribonucleoprotein (snRNP) particles required for splicing[10]. Within CBs, essential but frequently limited factors necessary for the generation of snRNAs and maturation of snRNPs are concentrated[11]. Interestingly, CBs are frequently in the proximity of highly expressed genes, presumably increasing the availability of snRNPs that facilitate splicing[12]. In *T. brucei*, we identified three discrete nuclear bodies, namely a CB, a spliced leader array body (SLAB), and a novel NUFIP splicing factor body.

Here, we show that these bodies, as well as the previously identified ESB[7], frequently associate with the active *VSG* ES, thus facilitating very high levels of VSG expression. We find that a feedback loop exists, whereby high levels of *trans*-splicing are required to maintain processive transcription elongation at the active ES. The congregation of a limited number of nuclear splicing factor bodies at the active ES to form an ES nuclear body assembly provides an appealing model for how the monoallelic expression of a single ES is maintained.

## Results

### Extremely high splicing levels facilitate ES transcription elongation.
In order to understand how vast amounts of *VSG* mRNA are produced from a single *VSG* gene, we compared rates of mRNA generation from different *T. brucei* loci. We estimated the number of mRNA molecules produced per hour from different genes, using available data on mRNA abundance and half-lives[13,14] (Fig. 1a and Supplementary Tables 1 and 2, see 'Methods' for the explanation of calculation). In BF *T. brucei*, an estimated 666 VSG mRNAs are generated per hour, compared with 1.3 mRNA per Pol II-transcribed gene, estimated by determining an average of all RNA polymerase II (Pol II) transcribed alleles. In comparison, in procyclic form (PF) *T. brucei*, the level of Pol I-transcribed EP and GPEET procyclin transcripts generated per hour is 30 to 54-fold higher respectively, than from an average Pol II-transcribed gene.

This phenomenal rate of VSG mRNA generation is supported by extraordinarily high rates of *trans*-splicing at the active ES, compared with the average Pol II-transcribed gene. We estimated that VSG pre-mRNA is spliced at a 512-fold higher rate, and an average ES mRNA at a ~40-fold higher rate than mRNA from the average Pol II-transcribed gene (Fig. 1b). The high level of splicing at the active ES is accompanied by an enrichment of U2 snRNA at this highly transcribed locus. We detected U2 snRNA using RNA-FISH throughout the BF *T. brucei* nucleus, as well as in an additional bright focus. This U2-positive focus colocalised with nascent RNA at the active ES, detected with a VSG pseudogene (VSGΨ) probe (Fig. 1c). The VSGΨ is unique to BES1, and produces an unstable transcript with a half-life of $1.63 \pm 0.65$ min, which does not appear to leave the nucleus[8].

We investigated the functional importance of splicing at the active ES by inhibiting it. We monitored nascent transcripts from *T. brucei* loci where 24× MS2 repeat containing constructs were integrated. The 24× MS2 repeats do not contain RNA processing signals, and the transcripts derived from them are not processed into mature mRNA, and are rapidly degraded. We have previously shown that MS2 repeat containing transcripts generated from the active ES are highly unstable, with a half-life of $1.75 \pm 0.62$ min[8]. These 24× MS2 repeat marked loci included the promoter and telomeric regions of the active BES1 (VSG221 ES), as well as the ribosomal DNA (rDNA) and tubulin loci (Supplementary Fig. 1a). We then transfected cells with anti-U2 snRNA Morpholinos, which have previously been shown to rapidly and specifically block *trans*-splicing in *T. brucei*[15].

Strikingly, this *trans*-splicing block resulted in a rapid reduction in cells with ES-derived nascent transcripts. This was more pronounced for transcripts from the ES telomere compared with the ES promoter (Fig. 1d and Supplementary Fig. 1b). This argues that although Pol I transcription still initiates when *trans*-splicing is blocked, elongation is partially inhibited. As BES1 extends for more than 60 kb[16], a transcript from the BES1 telomere could be expected to be affected more by a partial elongation block than one from the promoter region. This apparent block in transcription elongation is possibly caused by the accumulation of nascent unspliced transcript impacting on Pol I processivity. In contrast, rRNA or tubulin nascent transcripts did not decrease, indicating that a *trans*-splicing block specifically impacts on transcription elongation at the active ES.

In addition to using RNA-FISH, unprocessed or nascent transcripts from these loci were also quantitated with qPCR, using one primer annealing outside of the transcript splice/processing site and the other annealing within the transcript itself. This showed that a splicing block resulted in a reduction in nascent telomeric ES transcript (VSGΨ) to 13% normal (Supplementary Fig. 2a), while nascent RNA from the rDNA or actin loci was either marginally reduced or not affected by the splicing block. Transfection with random sequence Morpholinos did not result in transcript level changes (Supplementary Figs. 1c and 2b). These data indicated that although virtually all Pol II-transcribed loci in *T. brucei* require U2 snRNA for *trans*-splicing, the active BES is particularly dependent on it. This is presumably due to the much higher requirement for splicing (and therefore U2 snRNA) at the active BES compared with the average Pol II-transcribed locus (Fig. 1b). In addition, BF *T. brucei* is exquisitely sensitive to knockdown of *VSG* transcript, which results in a very rapid growth arrest[17].

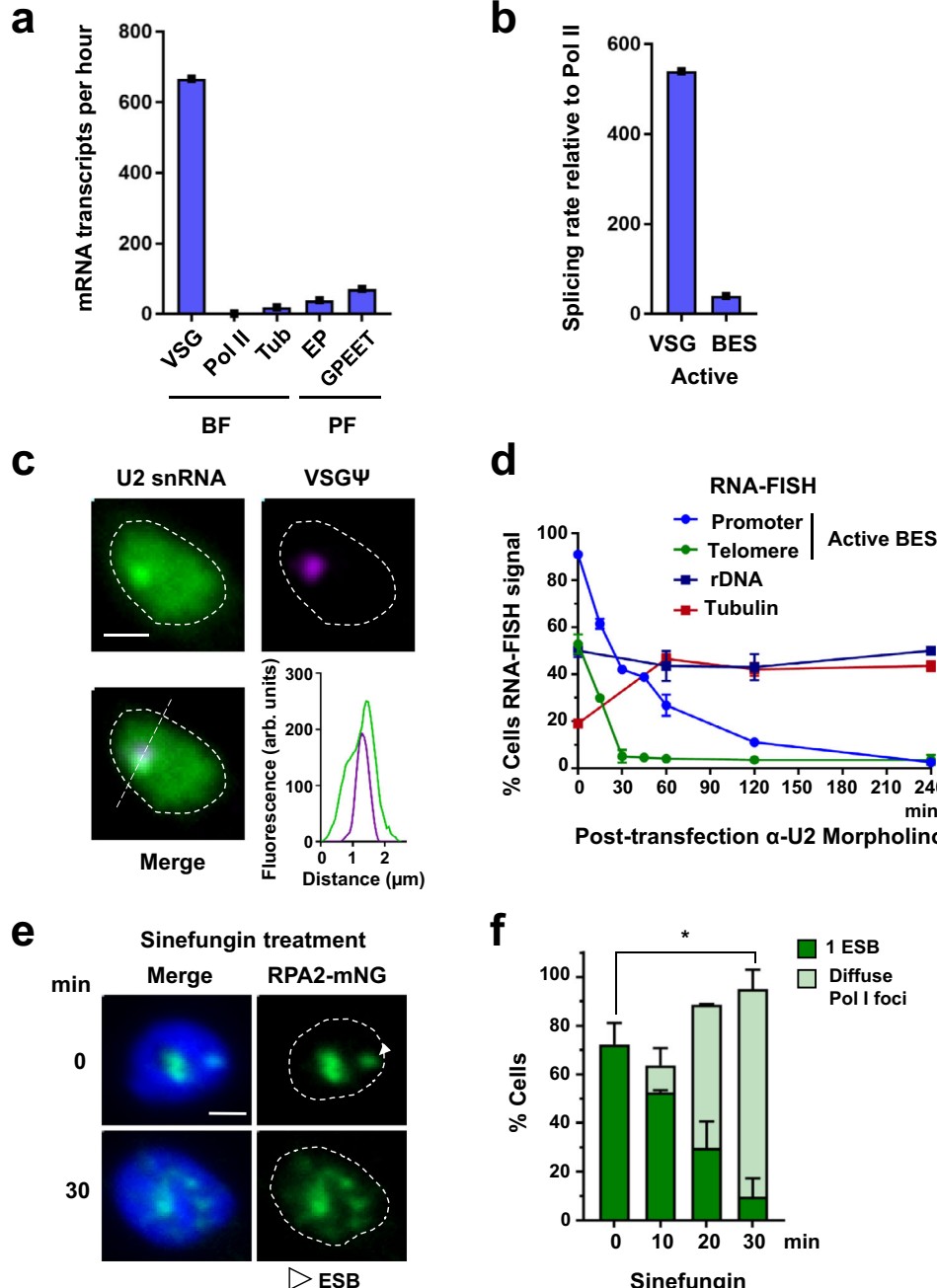

**Fig. 1 Extraordinarily high levels of splicing occur at the active VSG ES, which is necessary for its transcription. a** Estimated mRNA transcripts produced per hour from an active *VSG* gene, an average of all Pol II-transcribed genes (Pol II), or a tubulin gene (Tub) in bloodstream form (BF) *T. brucei*. This is compared with mRNA from a procyclin (EP or GPEET) gene in procyclic form (PF) *T. brucei*. **b** Estimated difference in splicing rate (fold difference) at the active *VSG* gene or an average ES gene (BES) compared with an average of all Pol II-transcribed genes. The data used to calculate the values presented in panels (**a**, **b**) are shown in Supplementary Tables 1 and 2. **c** Representative fluorescence microscopy images of RNA-FISH detecting the U2 snRNA (green) or VSG pseudogene (Ψ) transcript (magenta) specific for BES1 in BF *T. brucei* cells. The nucleus is indicated with a dashed line, scale bar = 1 μm. A line profile displaying the U2 snRNA and VSGΨ signal intensities is shown, with fluorescence indicated in arbitrary units (arb. units). **d** Quantitation of the percentage (%) of cells with MS2 RNA-FISH signal in cells with different genomic loci marked with MS2 repeats, at different time points in minutes (min) after transfection with anti-U2 snRNA Morpholinos. Minimally 300 G1 (1K1N) cells were counted per time point. Values shown are the average of three biological replicates, error bars indicate ± SD. **e** Fluorescence microscopy imaging of a BF *T. brucei* cell line expressing mNG::RPA2 (green) treated with sinefungin. Time is indicated in minutes (min). The nucleus (dashed line) is stained with DAPI (blue), and the ESB indicated with a white arrowhead. Scale bar = 1 μm. **f** Quantitation of the percentage of cells with an ESB in the cell line shown in (**e**) after treatment with sinefungin. Values do not add up to 100%, as not all cells had a detectable ESB. Average values for three biological replicates are shown. At least 200 G1 (1K1N) cells were counted per time point with error bars indicating ± SD. P values were determined using a two-tailed paired Student's *t* test. \*P = 0.0168. Source data are provided as a Source Data file.

This apparent perturbation of ES transcription elongation in BF *T. brucei* was also observed using the splicing inhibitor sinefungin (Supplementary Fig. 3a). Treatment resulted in a preferential rapid decrease in nascent transcripts from the active ES telomere compared with the promoter, similar to as observed after transfection with anti-U2 Morpholinos. Nascent transcripts derived from the rRNA or tubulin loci were still present after splicing inhibition. As these were sensitive to the transcription inhibitor Actinomycin D (Supplementary Fig. 3b), their presence after splicing inhibition was presumably a consequence of continuous transcription. As splicing inhibition resulted in a severe transcription defect at the active ES, we monitored the ESB. Normally $72 \pm 9\%$ cells have one ESB, however this decreased to $10 \pm 7\%$ ($P = 0.01$) after 30 min incubation with sinefungin, (Fig. 1e, f). Therefore, the extraordinarily high rates of splicing at the active ES appear necessary for processive Pol I transcription.

**Identification of a spliced leader array body (SLAB) specifically associating with the active ES.** We investigated if a nuclear body facilitating splicing was associated with the active ES. Cajal bodies (CBs) are nuclear bodies involved in the maturation of the small nuclear and nucleolar ribonucleoproteins (snRNPs and snoRNPs) necessary for processing pre-mRNAs and pre-rRNA[10]. Coilin is an essential Cajal body protein with limited sequence conservation across eukaryotes. However, CBs have not been defined in *T. brucei* and we could not identify coilin homologues in the *T. brucei* genome. We, therefore, interrogated TrypTag, a database with localisation patterns of PF *T. brucei* proteins[18], searching for potential CBs or related nuclear bodies, defined as one to four discrete nuclear foci. Screening 1478 nuclear proteins (using TryTrypDB release 44), revealed nine proteins with a CB-like localisation (Supplementary Table 3). Interestingly, VEX1 (VSG exclusion 1) and VEX2 were present, which associated with the ESB and are important for *VSG* monoallelic exclusion[19,20]. In addition, we identified SNAP3 (small nuclear RNA-activating protein 3). In kinetoplastid protozoa, SNAP3 is part of the SNAP transcription factor complex binding the SL-RNA gene arrays, and driving transcription of the SL RNAs required for *trans*-splicing[21]. We epitope-tagged different combinations of these proteins and determined their cellular localisation in BF *T. brucei*.

SNAP3 appeared to colocalise with VEX1 (in agreement with ref. [22]), as well as a previously unidentified protein, which we called Spliced Leader Array Protein 1 (SLAP1, Tb927.11.9950). These proteins colocalised in a structure which we called the Spliced Leader Array Body (SLAB) (Fig. 2a). Bioinformatic analysis of SLAP1 revealed disordered and coiled-coil domains (Fig. 2b). Homologues were only detectable in kinetoplastid protozoa, with no homology to proteins with known function (Supplementary Fig. 4a). Using SLAP1 as a marker, we found that the majority ($58 \pm 7\%$) of BF *T. brucei* cells in G1 (1K1N) had a single SLAB focus, with $42 \pm 7\%$ containing two (Fig. 2c). In G1 cells that contained two SLAB foci, these foci were usually separated with a mean distance of $870 \pm 345$ nm (Supplementary Fig. 4b). We determined the proximity of the SLAB components SNAP3, VEX1 and SLAP1 to the active ES using structured illumination super-resolution microscopy (SR-SIM) in cells where each of these SLAB components was epitope-tagged. The active ES was monitored using RNA-FISH detecting nascent RNA from the active BES1 (VSGΨ)[8]. These SLAB components frequently appeared associated with the active ES, with an average distance of $360 \pm 250$ nm (Fig. 2d, e). As the SLAB assembles at the SL arrays, our results agree with ref. [22], showing that the SL arrays associate with the active ES.

To determine if SLAB interaction with BES1 was a consequence of its activation state, we investigated its location relative to the two dynamically active ESs in double-expresser (DE) cells. In *T. brucei* DE KW01, cells rapidly switch back and forth between two unstably active ESs[8]. Cells therefore contain nascent transcripts from both ESs, or either ES individually. We epitope-tagged SLAP1 in *T. brucei* DE KW01, and performed RNA-FISH detecting BES1 and BES2 nascent RNA. This showed the SLAB associated with whichever ES was actively transcribed (Fig. 2f, g and Supplementary Fig. 4c). SLAB association with the active ES was independent of VEX1, as VEX1 knockdown did not affect its proximity to the active ES (Supplementary Fig. 4d, e).

**SLAP1 is a novel regulator of SL-RNA production and is essential for transcription of the active ES.** SLAP1 appears to be an essential gene (Supplementary Fig. 5a). As SLAP1 is colocalised with SNAP3, we investigated if its knockdown with RNAi disrupted SL-RNA production. We monitored the SL-RNA intron by RNA-FISH after induction of SLAP1 RNAi. The SL-RNA intron was undetectable after induction of SLAP1 RNAi for 16 h, suggesting that SLAP1 is essential for SL-RNA transcription and/or production (Fig. 3a). SLAP1 knockdown also led to an increase in actin nascent RNA, indicating that splicing had been inhibited (Fig. 3b). In addition, the SNAP3 signal became more diffuse, and there was an increase in cells with two foci rather than one (Supplementary Fig. 5b).

We had shown that splicing inhibition with Morpholinos or sinefungin disrupted transcription elongation at the active ES, and maintenance of the ESB. SLAP1 knockdown resulted in a similar phenotype. Using RNA-FISH and qPCR, we found that nascent transcript from the active ES telomere (VSGΨ) was significantly reduced after induction of SLAP1 RNAi (Fig. 3b–d). There was also ESB disruption, with the percentage of cells with an ESB declining from $75 \pm 8\%$ to $14 \pm 3\%$, after induction of SLAP1 RNAi for 16 h ($P = 0.0031$) (Fig. 3e, f). The SLAB remained intact after specific inhibition of Pol I transcription using the transcription inhibitor BMH-21[23] (Supplementary Fig. 5c, e). However, an increased number of cells had two SLAB foci rather than one (Supplementary Fig. 5b–d). As there are two SL-RNA arrays in *T. brucei*, this could indicate that active ES transcription helps drive colocalisation of these two SL gene arrays in the BF. SLAP1, therefore, appears to be a novel regulator of SL-RNA production. This continued SL-RNA synthesis is necessary for the *trans*-splicing which facilitates processive ES transcription.

**A novel NUFIP body associates with the active ES.** We also discovered a novel nuclear body in BF *T. brucei*, closely associating with the active ES. Using combined epitope tagging of proteins identified using TrypTag, we found five proteins that colocalised in this unique structure (Fig. 4). The conserved protein NUFIP (nuclear FMR1 interacting protein 1, Tb927.3.3740) was present, as well as the RBP34 RNA binding protein (Tb927.11.3340), the ZNHIT3 zinc finger protein (Tb927.11.2900) and two unknown hypothetical proteins with RNA-recognition motifs, which we called NufB1 (Tb927.4.3150) and NufB2 (Tb927.9.13970) (Fig. 4a). We called this nuclear structure the NUFIP body (NB), as NUFIP was the most highly conserved protein within it. We performed super-resolution microscopy analysis of BF *T. brucei* cell lines containing epitope-tagged copies of different combinations of NB components, and showed that they all colocalised with each other (Fig. 4b–d). Interestingly, higher magnifications indicated that the NB was further sub-compartmentalised into a "shell"-type structure (insets in Fig. 4b–d). The relatively spherical

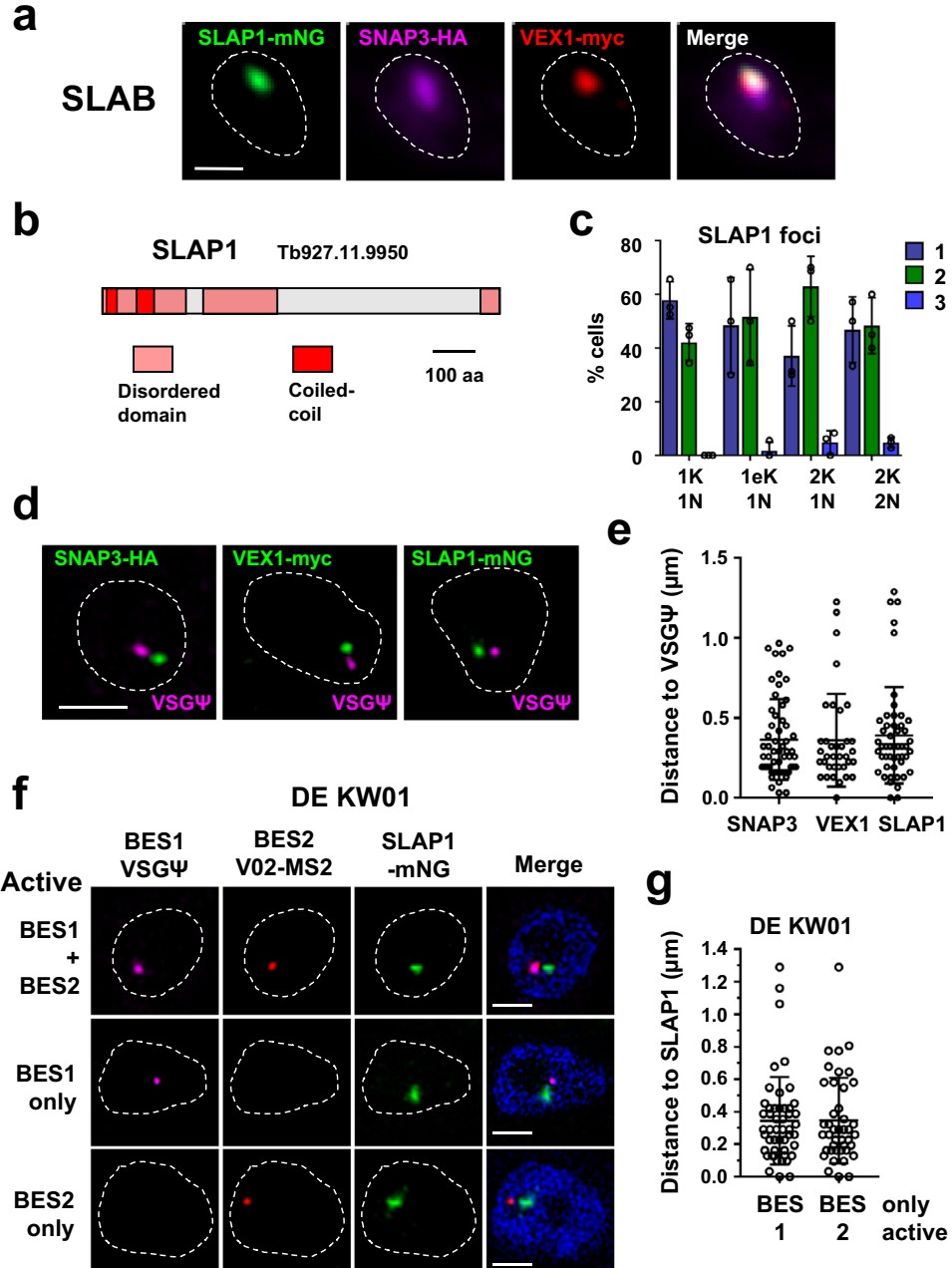

**Fig. 2 Identification of a Spliced Leader Array Body (SLAB) which frequently associates with the active ES. a** Fluorescence microscopy imaging of a triple epitope-tagged BF *T. brucei* cell line expressing the SLAB components SLAP1, SNAP3 and VEX1. Signals in the nucleus (dashed line) correspond to mNG::SLAP1 (SLAP-mNG, green), SNAP3::6xHA (SNAP3-HA, magenta) and VEX1::12xmyc (VEX1-myc, red). Scale bar = 1 μm. **b** Schematic of SLAP1 protein motifs identified using PFAM and disordered domains with PONDR. **c** Quantitation of the percentage (%) of BF *T. brucei* cells with different numbers of SLAP1 foci (1–3), visualised using an mNG::SLAP1 cell line. Cells at different cell cycle stages with various numbers of kinetoplasts (K), elongated kinetoplasts (eK) or nuclei (N) were analysed. Values are the averages of three biological replicates with error bars indicating ± SD. For 1K1N $n = 218$, 1eK1N $n = 57$, 2K1N $n = 38$ 2K2N $n = 34$. Values obtained from each replicate are shown with circles. **d** SR-SIM imaging of individually epitope-tagged BF *T. brucei* cell lines expressing the SLAB1 components SNAP3, VEX1 and SLAP1 in the nucleus (dashed line). Cell lines express: SNAP3::6xHA (SNAP3-HA, green), VEX1::12xmyc (VEX1-myc, green) or mNG::SLAP1 (SLAP1 mNG, green). RNA-FISH detects nascent transcript from the active BES1 using the VSG pseudogene (VSGΨ, magenta) probe. Scale bar = 1 μm. **e** Quantitation of distance between SNAP3, VEX1 or SLAP1 foci to the VSGΨ transcript using the cell lines shown in (**d**). Average values from minimally two biological replicates are shown with error bars = ± SD. For SNAP3 $n = 57$, VEX1 $n = 37$ and SLAP1 $n = 50$, only 1K1N cells were quantitated. **f** SR-SIM imaging of the double-expresser (DE) BF *T. brucei* line DE KW01-MS2-V02 in which both BES1 and BES2 are simultaneously unstably active. SLAP1 is epitope-tagged: mNG::SLAP1 (SLAP1 mNG, green). Nascent BES1 transcripts were visualised with RNA-FISH using a VSG pseudogene (VSGΨ, magenta) probe, and BES2 transcripts using MS2 repeats inserted within BES2 (V02-MS2, red). Nuclei (dashed lines) are stained with DAPI (blue), and represent cells with transcription of BES1 and BES2 simultaneously or individually. Scale bar = 1 μm. **g** Quantitation of distance of BES1 or BES2 transcript to SLAP1 in cells shown in (**f**), when transcription of either BES1 or BES2 occurs individually. Data are from two biological replicates with error bars = ± SD. For cells with BES1 signal only, $n = 42$. For cells with BES2 signal only, $n = 49$. Source data are provided as a Source Data file.

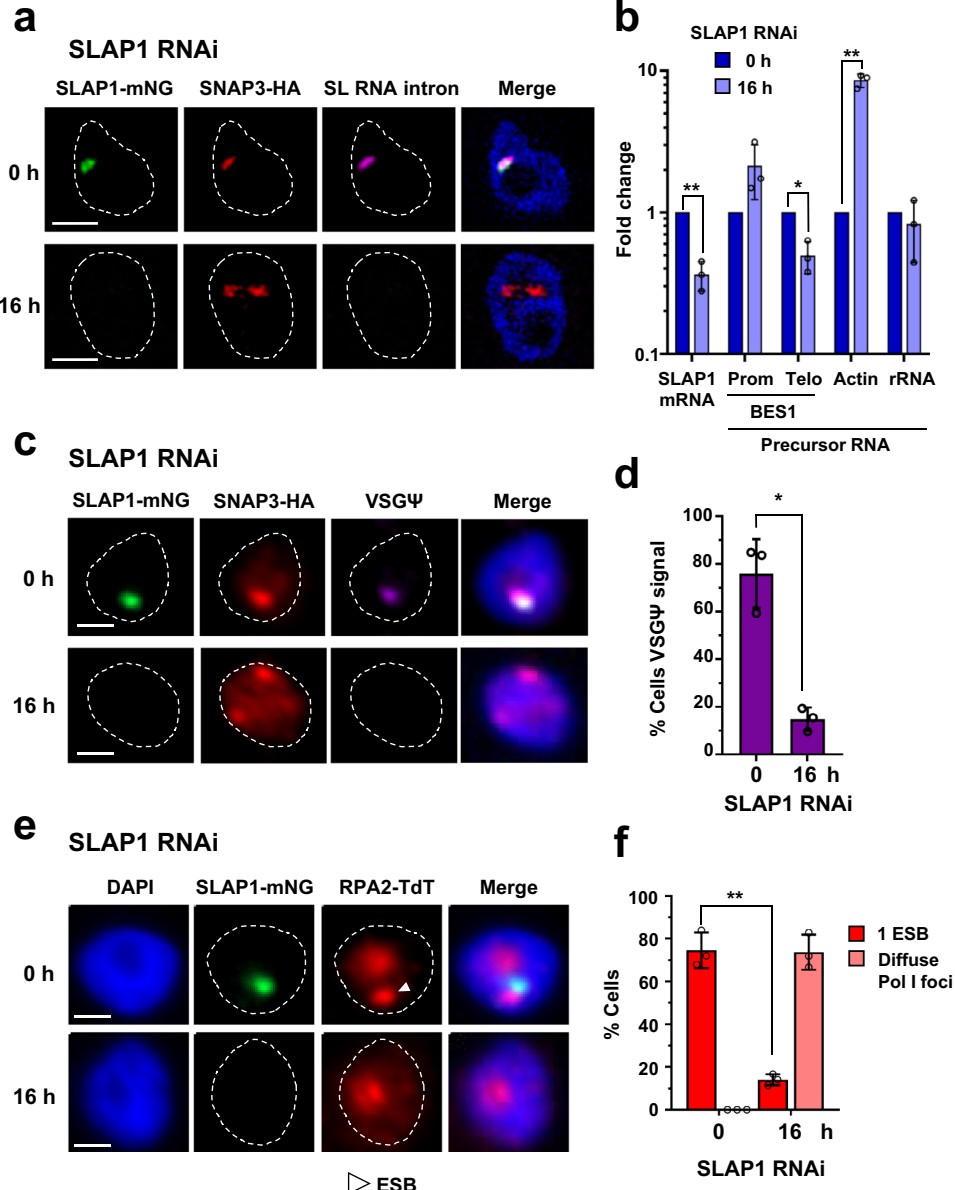

**Fig. 3 SLAP1 regulates SL-RNA production with knockdown resulting in an ES transcription block. a** SR-SIM imaging of a BF *T. brucei* cell line with nuclear signals corresponding to mNG::SLAP1 (SLAP1 mNG, green) or SNAP3::6xHA (SNAP3-HA, red), with the SL-RNA intron (magenta) visualised with RNA-FISH. SLAP1 RNAi was induced for the time in hours (h). **b**) RNA quantitation using qPCR from cells in (**a**) after induction of SLAP1 RNAi for the time in hours (h). Nascent RNA is from the promoter (Prom) or telomere (Telo) of BES1, or from the actin or rRNA loci. Average values are from three biological replicates with error bars indicating ± SD. **P ≤ 0.01; *P ≤ 0.05. For SLAP1 mRNA, P = 0.006, BES1 telomere pre-mRNA P = 0.02 and actin pre-mRNA P = 0.0051. **c** Fluorescence microscopy imaging of a BF *T. brucei* cell line, with mNG::SLAP1 (SLAP1 mNG, green) and SNAP3::6xHA (SNAP3-HA, red) signal shown, with VSGΨ transcript (magenta) detected with RNA-FISH. SLAP1 RNAi was induced for the time in hours (h). **d** Quantitation of the percentage (%) of cells in (**c**) with VSGΨ RNA-FISH signal after induction of SLAP1 RNAi for the time in hours (h). Values are the average of three biological replicates, with error bars indicating ± SD. At least 297 1K1N cells were counted for each condition. *P = 0.029. **e** Fluorescence microscopy imaging of a BF *T. brucei* cell line with a signal corresponding to mNG::SLAP1 (SLAP1-mNG, green) or TdT::RPA2 (RPA2-TdT, red), and a white arrowhead indicating the ESB. SLAP1 RNAi was induced for the time in hours (h). **f** Quantitation of the percentage (%) of cells in (**e**) with a discrete ESB after induction of SLAP1 RNAi for the time indicated in hours (h). Averages of three biological replicates are shown, with error bars indicating ± SD. At least 247 1K1N cells were counted for each condition. **P = 0.0031. In all microscopy images nuclei (dashed lines) are stained with DAPI (blue). On bar graphs, the values from individual replicates are shown with circles. All P values were determined using a two-tailed paired Student's *t* test. All scale bars = 1 μm. Source data are provided as a Source Data file.

structure of the NB, which is subdivided into subdomains, could indicate that it is formed through phase separation[24]. Most BF *T. brucei* cells had a NB, with 75 ± 2% of G1 cells having a single body and 19 ± 3% having two (Fig. 5a). In G1 cells that contained

two NUFIP bodies, these foci were normally separated with a mean distance of 885 ± 358 nm (Supplementary Fig. 8a).

To determine the possible function of the NB, we performed bioinformatic analysis on its components. NUFIP has been

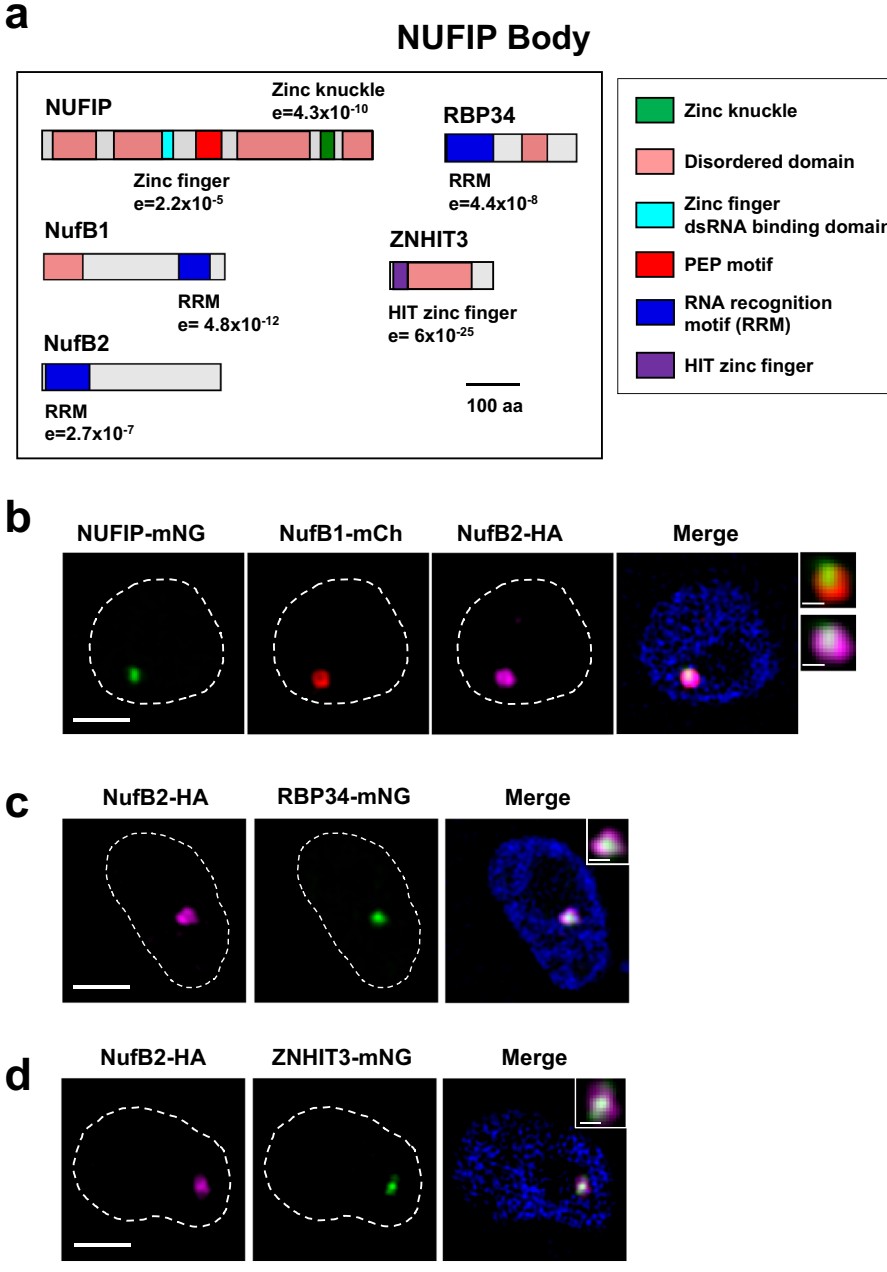

**Fig. 4 Identification of a novel NUFIP body. a** Schematic of NUFIP body components: NUFIP, RBP34, ZNHIT3 and the newly identified NufB1 and NufB2 proteins. Relevant protein domains and motifs were identified using PFAM, and disordered domains with PONDR (coloured boxes). **b** SR-SM imaging of a BF *T. brucei* cell line with three NUFIP components epitope-tagged. Signals in the nuclei (dashed lines) correspond to NUFIP::mNG (NUFIP-mNG, green), mCh::NufB1 (NufB1-mCH, red) and NufB2::6HA (NufB2-HA, magenta), with DNA stained with DAPI (blue). The insets show further magnification of the foci, illustrating the 'shell'-like structure. Scale bar = 1 μm for main images, and 200 nm for the insets. **c** As in (**b**), only showing SR-SM microscopy imaging of a BF cell line with two NUFIP body components epitope-tagged: NufB2::6HA (NufB2-HA, magenta) and RBP34::mNG (RBP34-mNG, green). **d** As in (**b**), but showing SR-SM microscopy imaging of a BF cell line with two NUFIP body components epitope-tagged: NufB2::6HA (NufB2-HA, magenta) and ZNHIT3::mNG (ZNHIT3-mNG, green).

shown to be involved in the assembly of U4 snRNPs in other organisms[25,26]. The PEP motif is the most highly conserved region of NUFIP and is found in NUFIP homologues across eukaryotes (Fig. 4a and Supplementary Fig. 6a, b). The PEP motif is important in mediating the interaction between NUFIP and the EDK motif in the CB protein Snu13[27], which is also conserved in Kinetoplastid protozoa (Supplementary Fig. 6c). Kinetoplastid NUFIP also contains two predicted RNA binding domains and several disordered regions (Fig. 4a and Supplementary Fig. 6d, e). One of these RNA binding domains is a zinc knuckle domain

which is typically found in splicing factors and other pre-mRNA processing proteins[28,29], and which appears to be unique for Kinetoplastid NUFIP (Supplementary Fig. 6a, d). In addition, we found that the HIT zinc finger protein (Tb927.11.2900) located in the NB (Fig. 4a, d) is a homologue of ZNHIT3 (Supplementary Fig. 7a). ZNHIT3 is well conserved across eukaryotes and has been shown to interact with NUFIP to mediate snRNP assembly[25,26].

We could not find homologues of known function to NufB1, NufB2 or RBP34, although each of these proteins contains a

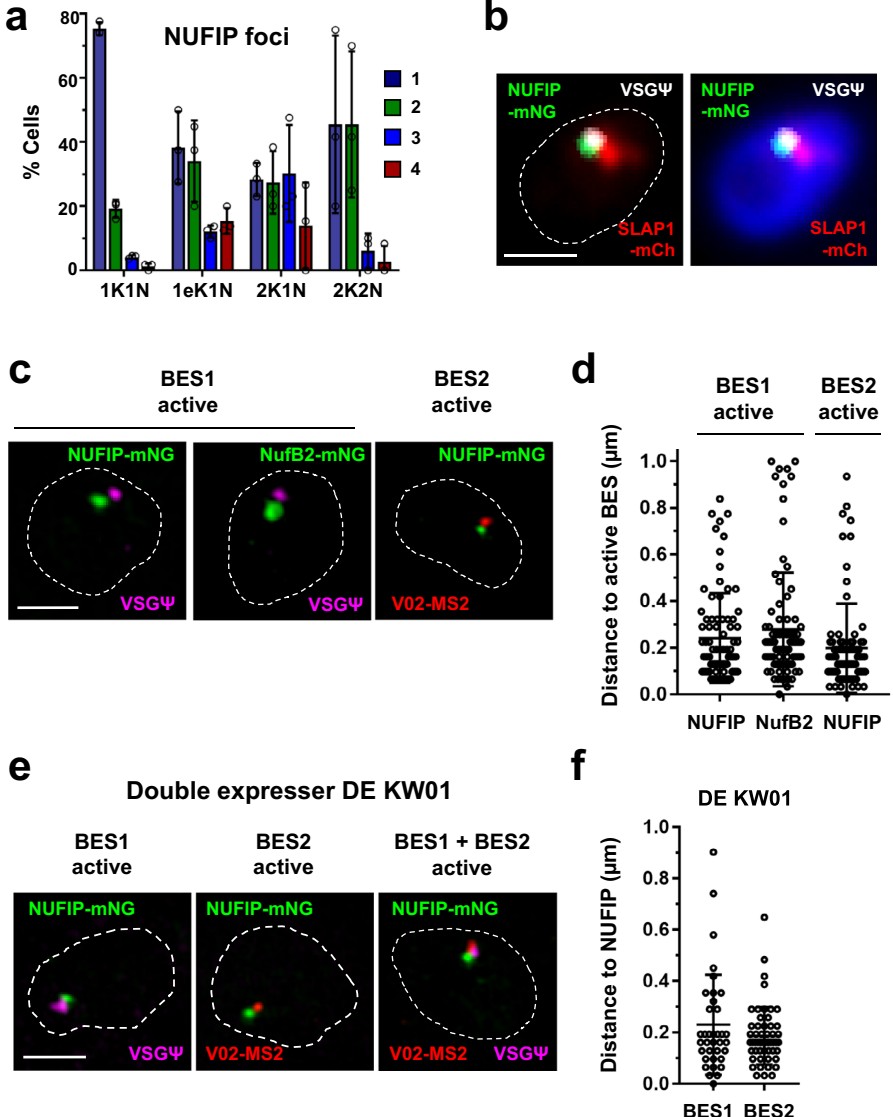

**Fig. 5 The NUFIP body associates with the active ES in BF *T. brucei*. a** Quantitation of the percentage (%) of BF *T. brucei* cells with different numbers of NUFIP foci (1–4) monitored using SR-SIM imaging. Different cell cycle stages are shown, with the number of kinetoplasts (K), elongated Kinetoplast (eK) or nuclei (N) indicated. Values are the averages from three biological replicates with error bars indicating ± SD. The values from individual replicates are shown with circles. For 1K1N $n = 333$, 1eK1N $n = 76$, 2K1N $n = 49$ and 2K2N $n = 38$. **b** Widefield microscopy imaging of a BF *T. brucei* cell line with the SLAB and NUFIP bodies visualised, combined with RNA-FISH detecting BES1 nascent RNA using a VSG pseudogene probe (VSGΨ, white). This line expresses mCh::SLAP1 (SLAP1-mCh, red) and NUFIP::mNG (NUFIP-mNG, green). The nucleus (dashed line) is stained with DAPI (blue). Scale bar = 1 μm. **c** SR-SIM imaging of a nucleus (dashed line) from BES1 expressing BF *T. brucei* which has the NUFIP body components NUFIP (NUFIP-mNG, green) or NufB2 (BUFB2-mNG, green) epitope-tagged with mNeon green. BES1 nascent transcript is detected with a VSG pseudogene (VSGΨ, magenta) probe. Alternatively, a BES2 expressing *T. brucei* line containing a MS2 repeat containing construct in BES2 was analysed which also had NUFIP epitope-tagged (NUFIP-mNG, green). BES2 nascent transcript were visualised with an MS2 probe (V02-MS2, red). Scale bar = 1 μm. **d** Quantitation of the distance between BES1 or BES2 nascent RNA and NUFIP or NufB2 using the cells in (**c**). Values are the averages from at least two biological replicates with error bars indicating ±SD. For active BES1 to NUFIP $n = 81$, active BES1 to NufB2 $n = 89$ and active BES2 to NUFIP $n = 79$. **e** SR-SIM imaging of combined RNA-FISH detecting nascent BES1 RNA (VSGΨ, magenta) or BES2 RNA (V02-MS2, red) with NUFIP::mNG (NUFIP-mNG, green). This was in BF *T. brucei* double-expresser cell line DE KW01-MS2-V02 where both BES1 and BES2 are unstably transcriptionally active. Scale bar = 1 μm. **f** Quantitation of the distance from BES1 or BES2 nascent RNA to the NUFIP body using cells from (**e**) in which RNA from either BES1 or BES2 is detectable. Values are the averages from two biological replicates, with error bars showing ±SD. For BES1 to NUFIP $n = 54$, for BES2 to NUFIP $n = 36$. Source data are provided as a Source Data file.

predicted RNA-recognition motif (Fig. 4a and Supplementary Fig. 7b, c). In *T. brucei* it has previously been found that NufB1, NufB2, RBP34, NUFIP and ZNHIT3 all associate with kinetochore proteins, although the role of RNA processing in chromosome segregation is unclear[30,31]. Interestingly, NufB1, NufB2, RBP34 and NUFIP have previously been found to interact with CRK9 in *T. brucei*[32], which is involved in methylation of the

SL-RNA cap, an RNA modification essential for *trans*-splicing[32]. These data argue that the possible function of the NB is that of a splicing factor body involved in snRNP assembly and/or SL-RNA modification.

We determined the position of the NB relative to the active ES using RNA-FISH and super-resolution microscopy, and found that it, as well as the SLAB, both invariably colocalised with the

nascent transcript from the active ES (VSGΨ) (Fig. 5b). The NB was in close proximity to the active BES1, on average $240 \pm 190$ nm (Fig. 5c, d). In cells where BES1 was silent and BES2 was active, the NB closely associated with the active BES2, on average within $200 \pm 190$ nm (Fig. 5c, d). The majority of double-expresser DE KW01 cells in G1 had a single NB ($77 \pm 7.5\%$) (Supplementary Fig. 8c), indicating that the dynamically active ESs in this cell line share a single NB as well as a single ESB[8]. We determined the nuclear positioning of the NB with respect to the dynamically active ESs in DE KW01 cells using RNA-FISH detecting nascent transcripts from BES1 and BES2, and showed that this was determined by which ES was active (Fig. 5e, f and Supplementary Fig. 8a). The NB was not sensitive to inhibition of Pol I transcription using BMH-21 (Supplementary Fig. 9a, b), indicating that it is distinct from the Pol I-enriched ESB. In summary, our data describe a novel NUFIP body, which has a possible function in snRNP assembly and/or SL-RNA modification, and which frequently associates with the active ES in BF *T. brucei*.

**The Cajal, SLAB and NUFIP bodies all frequently interact with the active ES.** Some of the proteins identified in TrypTag appeared localised in both nucleoli as well as CB-like foci. *T. brucei* appears to have three fibrillarin proteins. The fibrillarin we designated as fibrillarin2 (Tb927.10.14750), as well as the conserved CB protein NOP56 (Tb927.8.3750) were present in both nucleoli and CBs, presumably as they have a dual function in both rRNA and snRNA modification[33]. We epitope-tagged NOP56 and fibrillarin2 in BF *T. brucei*, and found that both proteins colocalised in both nucleoli and extra-nucleolar foci (Fig. 6a, b). The extra-nucleolar focus did not colocalise with the nucleolar marker L1C6 (Fig. 6c). As both NOP56 and fibrillarin are considered to be markers for the CB, these extra-nucleolar foci in *T. brucei* appear to be canonical CBs. In BF *T. brucei* a CB was detectable in $26 \pm 5\%$ of cells in G1 (Fig. 6b). Using RNA-FISH detecting nascent transcript from BES1 (VSGΨ), we found that if a CB was present, this was frequently near the active ES, on average at $410 \pm 240$ nm (Fig. 6d, e). In general, the CB was not as close to the active ES as the SLAB and NB, which were usually ~360 nm and 240 nm, respectively, from the active ES.

We next determined the relative proximities of these different nuclear bodies to each other in BF *T. brucei*, using cell lines where multiple nuclear bodies could be visualised simultaneously. Using super-resolution microscopy, we could clearly visualise these different nuclear bodies, which were usually within 500 nm of each other (Fig. 7). Example images of different configurations of these bodies are shown (Supplementary Fig. 10). The ESB, SLAB and NB were all within 500 nm of each other in 59% of the cells (Fig. 7a, b and Supplementary Fig. 10a). The ESB was within 500 nm of only a SLAB within 77% of the cells, and only a NB within 75% of the cells. Similarly, a CB, SLAB and NB were all within 500 nm of each other in 44% of the cells, with either a CB or a SLAB within 500 nm of the NB in 63% of the cells (Fig. 7c, d and Supplementary Fig. 10b). We also visualised the ESB using VEX2[20], which confirmed our results using Pol I as a marker. Here, the ESB was within 500 nm of the NB in 89% of the cells, and the SLAB in 52% of the cells (Supplementary Fig. 11).

We do not think that the frequent localisation of the CB, ESB, NB and SLAB within 500 nm of each other is fortuitous. The CB, ESB, NB and SLAB have relatively similar average diameters of around 235 nm, 287 nm, 222 nm or 317 nm, respectively (Supplementary Fig. 10c). We have previously determined the proximity of the BES1 and the unrelated tubulin locus in BF *T. brucei*[8], and showed that these two loci were separated from each other with an average distance of $880 \pm 480$ nm and located

within 500 nm of each other only 17% of the time. These data support the hypothesis that these nuclear bodies preferentially associate with the active ES.

In BF *T. brucei*, a highly SUMOylated focus (HSF) has previously been shown to be partially colocalised with the ESB[34]. We determined the location of the HSF relative to the different BF *T. brucei* nuclear bodies (Fig. 8). We found that, as previously documented, there is a significant overlap between the ESB as visualised with Pol I (RPA2-Halo) and the HSF (Fig. 8a). The mean distance between the centre of these two foci was $110 \pm 63$ nm, which was slightly further than the centre of the VEX2 focus to the ESB ($71 \pm 54$ nm) (Fig. 8a, c). Interestingly, using combined immunofluorescence/RNA-FISH experiments, we found that the HSF was in closer proximity to nascent RNA derived from the active ES promoter compared with the active ES telomere ($113 \pm 76$ nm versus $248 \pm 108$ nm, respectively) (Fig. 8b, c). In contrast, the centre of the ESB was almost equidistant to both active ES promoter and telomere transcripts ($128 \pm 88$ nm and $151 \pm 99$ nm, respectively). The HSF did not appear to overlap with the Cajal body, SLAB or NUFIP body (Fig. 8d, e) with mean distances of $490 \pm 301$ nm, $317 \pm 295$ nm and $279 \pm 251$ nm, respectively (Fig. 8f). These data collectively suggest that the HSF is tightly associated with the ESB, and is particularly close to the active ES promoter, possibly as the active ES promoter region is highly enriched for SUMOylated proteins[34].

As the NB and SLAB nuclear positioning seemed to be determined by which ES was active, we hypothesised that active ES transcription helped bring these nuclear bodies together. To test this, we determined the distance between the SLAB and the NB after blocking Pol I transcription with BMH-21. We found that there was a slight but significant increase in the distance between the SLAB and NB, from $435 \pm 357$ nm in untreated cells to $608 \pm 488$ nm after 4 h of treatment with BMH-21 ($P = 0.043$, Supplementary Fig. 9c, d). As BMH-21 is a reversible inhibitor, we restored Pol I transcription by washing it out[23]. After removal of BMH-21 for 30 min, the distance between the NB and the SLAB decreased to $521 \pm 368$ nm (Supplementary Fig. 9d). This indicates that Pol I transcription of the active ES is partly responsible for the close proximity of the NB to the SLAB in BF *T. brucei*. Further experiments will be needed to fully determine how these nuclear bodies are recruited to the active ES. However, collectively, these results show that in BF *T. brucei* the ESB, NUFIP, SLAB and Cajal bodies frequently associate with the active ES, forming an ES nuclear body assembly.

**PF *T. brucei* contains the SLAB, NUFIP and Cajal bodies, which do not appear to associate with a Pol I-transcribed procyclin locus.** The SLAB, NUFIP and Cajal bodies of BF *T. brucei* were identified using proteins forming discrete nuclear foci in PF *T. brucei* in the TrypTag database. However, it was unclear if these proteins were colocalised in the same manner in both life-cycle stages. To address this, PF *T. brucei* cell lines were generated in which different components of the SLAB, NB and CB were epitope-tagged in the same cell. Microscopy analysis of these cell lines revealed that the SLAB, NB and CB were also present in PF *T. brucei* (Fig. 9a). Both PF and BF *T. brucei* in G1 had a similar percentage of cells containing a SLAB or CB (Supplementary Fig. 12a, b). However, a NUFIP focus was visible in fewer PF cells ($45 \pm 6\%$) compared with the BF ($75 \pm 2\%$) (Supplementary Fig. 12c and Fig. 5a).

In PF *T. brucei*, the GPEET and EP procyclin genes encode surface proteins that replace VSG and are also transcribed by Pol I. However, there is no ESB in PF *T. brucei*. As the NB and SLAB are in close proximity to the active ES in BF *T. brucei*, we investigated if in PF *T. brucei* they were also associated with the

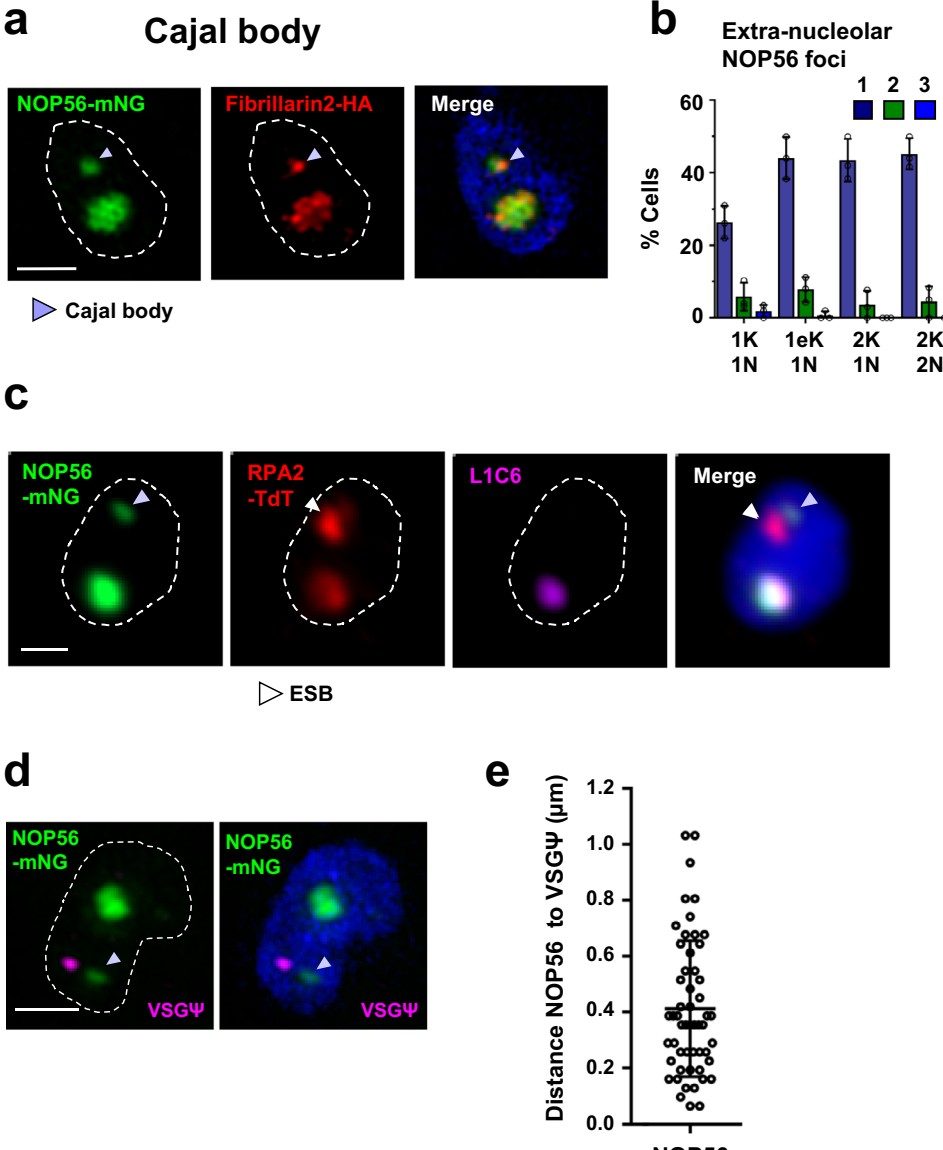

**Fig. 6 A subset of BF *T. brucei* cells have a Cajal body that associates with the active ES. a** SR-SIM imaging of a BF *T. brucei* cell line expressing two conserved Cajal body markers which are epitope-tagged: mNG::NOP56 (NOP56-mNG, green) and 3xHA::fibrillarin2 (fibrillarin2-HA, red). The Cajal body is indicated with a light-blue arrowhead. **b** Quantitation of the percentage (%) of BF *T. brucei* cells with 1–3 Cajal bodies, as determined by fluorescence microscopy imaging of a cell line expressing mNG::NOP56. Cells throughout the cell cycle containing different numbers of kinetoplasts (K), elongated kinetoplasts (eK) or nuclei (N) were monitored. Average values are shown from three biological replicates, with error bars, indicating ±SD. The values from individual replicates are shown with circles. For 1K1N $n = 284$, 1eK1N $n = 120$, 2K1N $n = 40$ and 2K2N $n = 30$. **c** Fluorescence microscopy imaging of a cell line expressing the Cajal body component mNG::NOP56 (NOP56-mNG, green) and the Pol I subunit TdT::RPA2 (RPA2-TdT, red). The nucleolus is visualised with an anti-L1C6 antibody (magenta), detecting an uncharacterised nucleolar protein. The Cajal body is indicated with a light-blue arrowhead and the ESB with a white arrowhead. **d** SR-SIM imaging of a cell line with epitope-tagged NOP56: mNG::NOP56 (NOP56-mNG, green) combined with RNA-FISH detecting nascent RNA from the active BES1 using a VSG pseudogene (VSGΨ, magenta) probe. **e** Quantitation of the distance between the Cajal body (NOP56) with the nascent transcript from BES1 (VSGΨ) using the cell line from (**d**). The data are from two biological replicates, ($N = 54$). Error bars indicate ±SD. For all microscopy images, nuclei are indicated with dashed lines, DNA is stained with DAPI (blue), and scale bars = 1 μm. Source data are provided as a Source Data file.

procyclin locus. We inserted MS2 repeats immediately downstream of an EP1 procyclin gene promoter in PF *T. brucei*, and monitored nascent RNA with RNA-FISH, in cells where NUFIP or SLAP1 were also epitope-tagged (Fig. 9b). The NB and SLAB did not appear obviously associated with the transcribed EP1 procyclin locus and were at an approximate distance of 800 ± 470 nm or 700 ± 360 nm, respectively (Fig. 9c). It is

therefore likely that the frequent proximity of the SLAB, NUFIP and Cajal bodies to the active ES in BF *T. brucei*, facilitates the extraordinarily high demand for splicing at this locus. We estimate that procyclin mRNA is generated at a rate of 71 or 39 transcripts per hour for GPEET and EP procyclin, respectively, compared with 666 mRNA transcripts per hour for VSG (Fig. 1a). Possibly these lower rates are not high enough to require nuclear

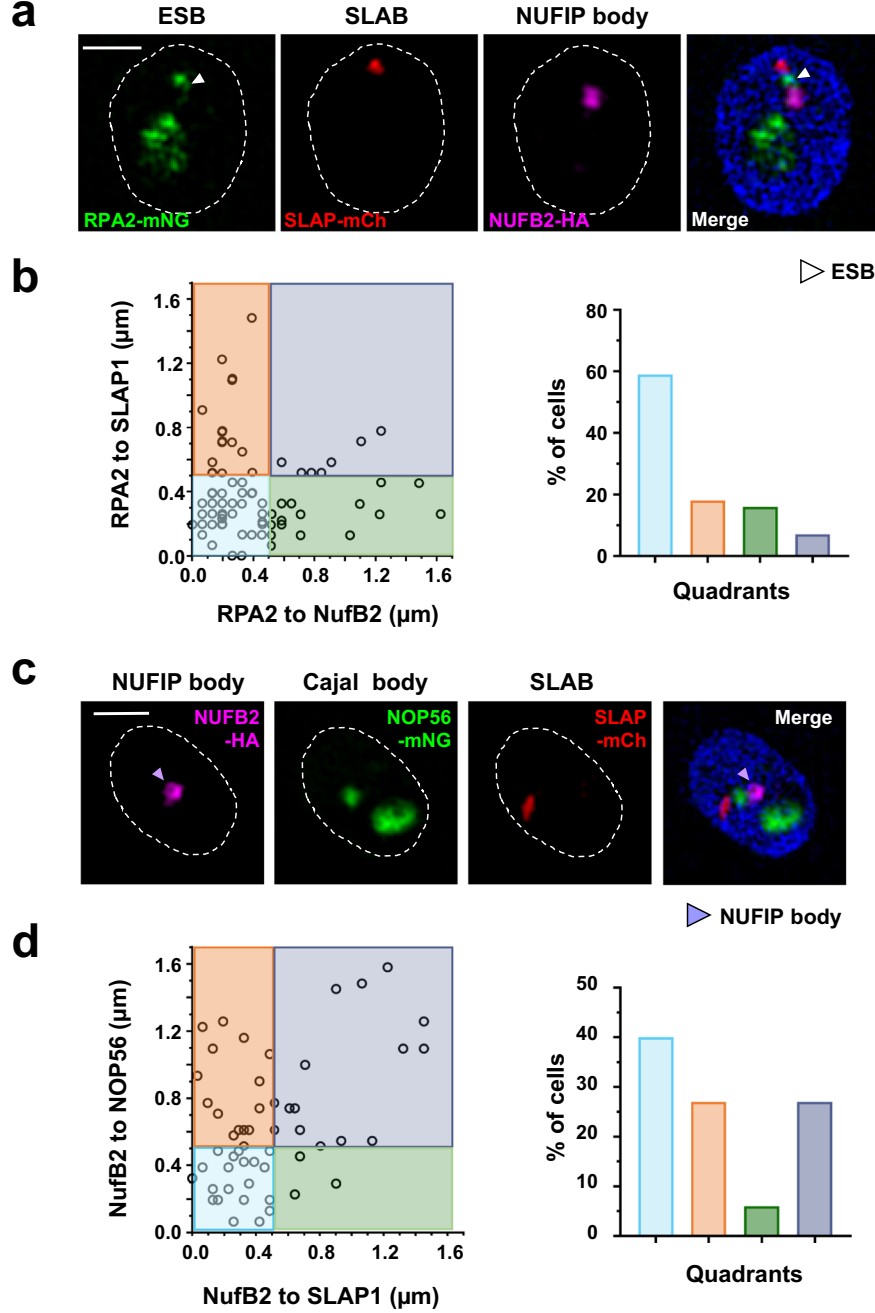

**Fig. 7 An assembly of nuclear bodies associates with the active ES in BF *T. brucei*. a** Simultaneous visualisation of the ESB, SLAB and NUFIP bodies using SR-SIM imaging of a triple epitope-tagged BF *T. brucei* cell line. This cell line expresses mNG::RPA2 (RPA2-mNG, green), mCh::SLAP1 (SLAP-mCh, red) and NufB2::6HA (NufB2-HA, magenta). The ESB is indicated with a white arrowhead within a nucleus (dashed line) stained with DAPI (blue). Scale bar = 1 µm. **b** Quantitation of the distance between the ESB (visualised with RPA2) and the SLAB (visualised with SLAP1) compared with the distance between the ESB and the NUFIP body (visualised with NufB2). Data was of the cell line shown in (**a**) using SR-SIM imaging. The scatter plot is divided into four quadrants which are coloured according to whether the different bodies in the cells are within 500 nm of each other. Quantitation of the percentage (%) of cells within each coloured quadrant is on the right. Example images of cells from each quadrant are shown in Supplemental Fig. 10. $N$ = 110 cells in G1 (1K1N) from two biological replicates. **c** Simultaneous visualisation of the Cajal, SLAB and NUFIP bodies using SR-SIM imaging of a triple epitope-tagged cell line. This line expresses mNG::NOP56 (NOP56-mNG, green), mCh::SLAP1 (SLAP-mCh, red) and NufB2::6HA (NufB2-HA, magenta). The NUFIP body is indicated with a light-purple arrowhead within a nucleus (dashed line) stained with DAPI (blue). Scale bar = 1 µm. **d** Quantitation of the distance of the Cajal body (CB) (visualised with NOP56) and the NUFIP body (visualised with NufB2) compared with the distance between the SLAB (visualised with SLAP1) and the Cajal body. Data were from the cell line shown in (**c**) using SR-SIM imaging. The scatter plot is divided into quadrants, which are coloured differently according to whether the different bodies in the cells are within 500 nm of each other. Example images of cells from each quadrant are shown in Supplemental Fig. 10. $N$ = 55 cells in G1 (1K1N) from two biological replicates. Source data are provided as a Source Data file.

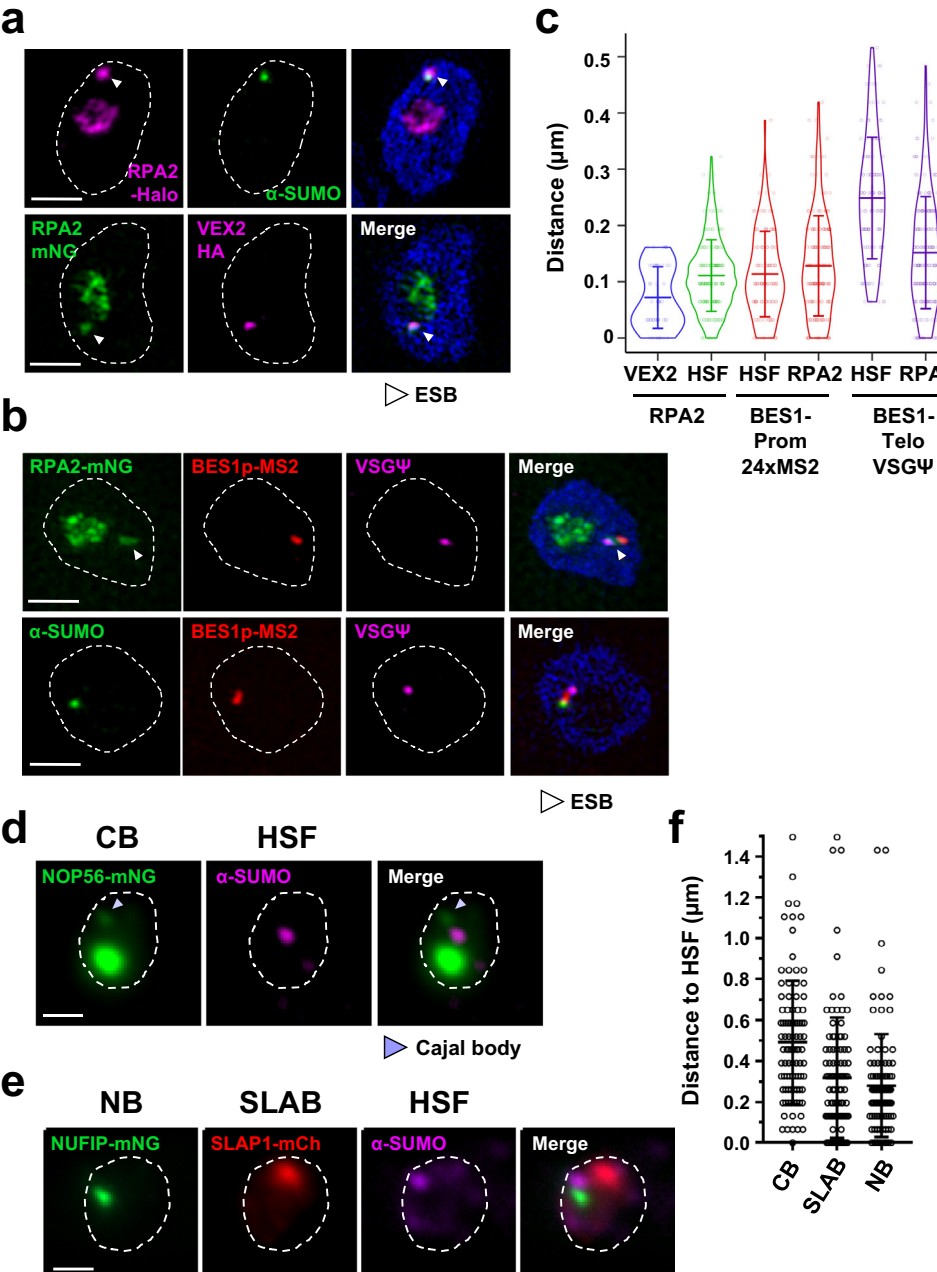

**Fig. 8 The highly SUMOylated focus is in close proximity to the ESB. a** SR-SIM imaging of the ESB (Halo::RPA2, magenta) with the highly SUMOylated focus (HSF; α-SUMO antibody, green) (upper panel) or the ESB (mNG::RPA2, green) with VEX2::6HA (magenta) (lower panel) in BF *T. brucei*. The ESB is indicated with a white arrowhead. **b** Combined RNA-FISH detecting nascent transcripts from the active BES1 promoter (BES1-Prom-24xMS2, red) and the active BES1 telomere (VSGΨ, magenta) with visualisation of the ESB (mNG::RPA2, green; upper panel) or the HSF (α-SUMO antibody, green; lower panel) using SR-SIM imaging. The ESB is indicated with a white arrowhead. **c** Violin plot showing quantitation of the distances of the ESB (RPA2) to either VEX2 or the HSF using microscopy images shown in (**a**) or the distances of nascent RNA from either the active BES1 promoter or telomere to either the HSF or ESB (RPA2) using microscopy images shown in (**b**). The number of 1K1N cells counted for each quantitation was VEX2-RPA2 *n* = 52, HSF-RPA2 *n* = 184, BES1 RNA-HSF *n* = 120, BES1 RNA-RPA2 *n* = 189. Cells with BES1 transcripts were only quantitated when both transcripts (promoter and telomere) were present. Measurements were taken from minimally two biological replicates. The broad lines indicate the mean with error bars indicating ±SD. **d** Immunofluorescence microscopy imaging of the Cajal body (CB) visualised with NOP56 (mNG::NOP56, green) and the HSF (α-SUMO antibody, magenta) in BF *T. brucei*. The Cajal body is indicated with a light-blue arrowhead. **e** Simultaneous imaging of the NUFIP body (NB) (NUFIP-mNG, green), SLAB (SLAP1-mCh, red) and HSF (α-SUMO antibody, magenta) in BF *T. brucei* using fluorescence microscopy. **f** Quantitation of the distance between the HSF and the Cajal body, SLAB or NUFIP body using the microscopy images shown in (**d**, **e**). Data were collected from 1K1N cells with minimally *N* = 97 from two biological replicates. Error bars indicate ±SD. For all microscopy images, nuclei are indicated with dashed lines, DNA is stained with DAPI (blue) and scale bars = 1 μm. Source data are provided as a Source Data file.

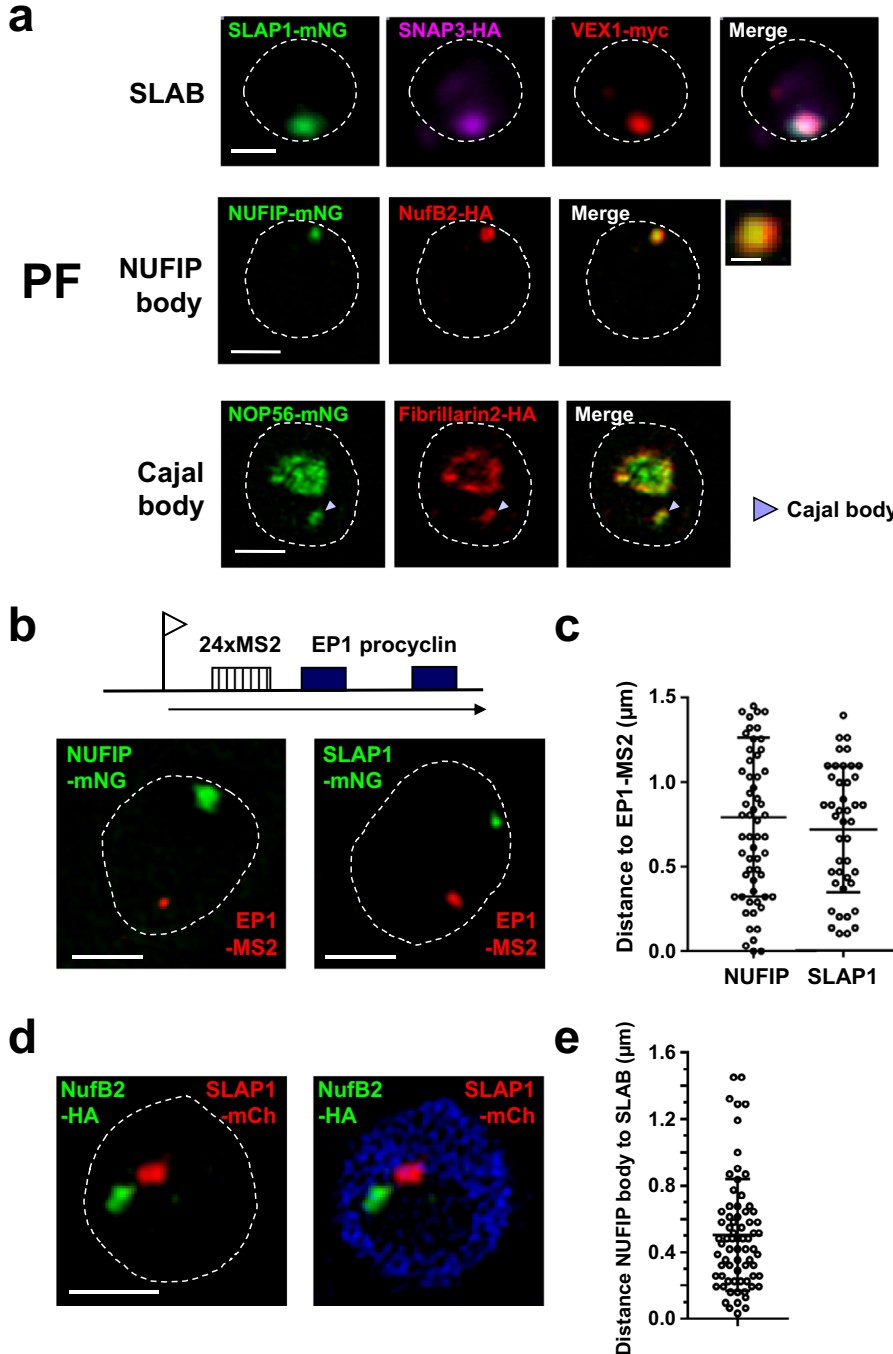

**Fig. 9 The SLAB, NUFIP and Cajal bodies are present in PF *T. brucei*, with the SLAB and NUFIP bodies frequently associated with each other, but not with the EP1 procyclin locus. a** The SLAB was investigated using microscopy imaging of a PF *T. brucei* line expressing the epitope-tagged SLAB components mNG::SLAP1 (SLAP-mNG, green), SNAP3::6HA (SNAP3-HA, magenta) and VEX1::12myc (VEX1-myc, red). The NUFIP body was investigated using SR-SIM imaging of a PF cell line expressing epitope-tagged NUFIP body components NUFIP::mNG (NUFIP-mNG, green) and NufB2::6HA (NufB2-HA, red). The inset image shows a magnification of the NUFIP body. The Cajal body was investigated with a PF *T. brucei* line expressing Cajal body components mNG::NOP56 (NOP56-mNG, green) and 3xHA::fibrillarin2 (fibrillarin2-HA, red). The arrowhead indicates the Cajal body. **b** SR-SIM imaging of RNA-FISH visualising nascent MS2 transcripts expressed from 24 MS2 repeats inserted in the EP1 procyclin locus (EP1-MS2, red) combined with imaging of NUFIP::mNG (NUFIP-mNG) or mNG::SLAP1 (SLAP-mNG, green). A schematic of the EP1 procyclin locus containing MS2 repeats (hatched box) is shown above, with the promoter indicated with a flag and transcription with an arrow. **c** Quantitation of the distance between the NUFIP body (NUFIP) or SLAB (SLAP1) to nascent MS2 transcripts expressed from the EP1 procyclin locus using the PF cells in (**b**). Data were from two biological replicates with error bars indicating ±SD. For EP1-MS2 to SLAP1 *n* = 43, for EP1-MS2 to NUFIP *n* = 63. **d** SR-SIM imaging of a PF *T. brucei* line expressing epitope-tagged components for the NUFIP: NufB2::6xHA (NufB2-HA, green) or SLAB: mCh::SLAP1 (SLAP1-mCh, red) bodies. **e** Quantitation of the distance between the NUFIP and SLAB bodies in PF *T. brucei* using the cell line in (**d**). Data were from two biological replicates with error bars indicating ±SD, *n* = 72. In all microscopy images, nuclei are indicated with dashed lines and scale bar = 1 μm for main images and 200 nm for the inset. Source data are provided as a Source Data file.

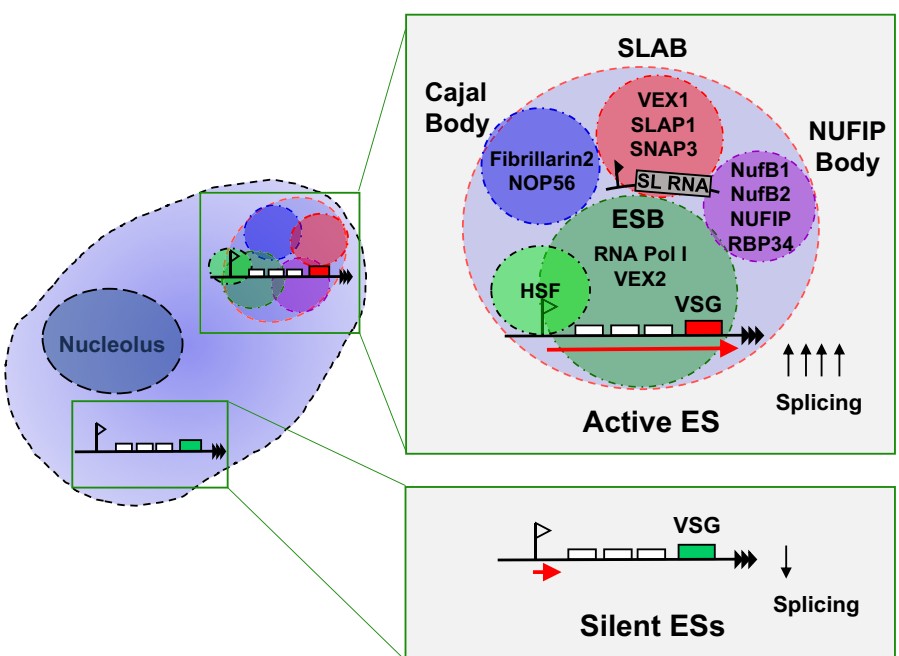

**Fig. 10 Model of the architecture of the Expression Site nuclear body assembly in BF *T. brucei*.** Schematic representation of a BF *T. brucei* nucleus (light blue) with the nucleolus indicated (dark blue). Active and silent ESs are expanded in the green boxes. The active ES is associated with four discrete nuclear bodies: The ESB (green circle), the Cajal body (dark blue circle), the SLAB (red circle) and the NUFIP body (purple circle). The ESB contains Pol I and VEX2, and shows partial overlap with a highly SUMOylated focus (SUMO). The SLAB closely associates with the spliced leader (SL) RNA gene locus. The ESs contain various genes (white boxes) in addition to the telomeric VSG gene, with the promoters indicated with flags, transcription with a red arrow and telomere repeats with black triangles. These four nuclear bodies appear to often cluster in a single nuclear compartment which forms an ES nuclear body assembly (large light-blue circle with red dashed outline) satisfying the high demand for splicing at the active ES. Silent ESs are not associated with this ES nuclear body assembly and the low amount of nascent RNA generated from these ESs is not spliced efficiently.

splicing factor bodies to be in the immediate vicinity of the procyclin genes.

However, the NB and SLAB in PF *T. brucei* were frequently close to one another, on average 503 ± 334 nm (Fig. 9a, d, e), similar to what we observed in BF *T. brucei* (470 ± 362 nm) (Fig. 7d). This could indicate that there is the transfer of components between these two splicing related compartments in both life-cycle stages. We, therefore, propose a model whereby in BF *T. brucei*, four nuclear bodies frequently associate with the active, but not the silent ESs (Fig. 10). These include the Pol I containing ESB, as well as the SLAB, NUFIP and Cajal bodies. In contrast, in PF *T. brucei* we have not found evidence for assembly of all of these bodies at a transcriptionally active procyclin locus, presumably due to the lower localised demand for splicing.

**Discussion**

Here, we identify three nuclear bodies (SLAB, NUFIP and Cajal) in both PF and BF *T. brucei*, which are apparently involved in the production of essential splicing components. The NUFIP body (NB) appears to be a novel nuclear compartment involved in snRNP assembly and/or SL-RNA modification. The SLAB is involved in the production of SL RNA. Despite the need for splicing factors throughout the nucleus, in BF *T. brucei*, these nuclear bodies are all associated with the active *VSG* ES. This results in an assembly of nuclear bodies associating with the ESB[7]. A single *VSG* gene produces an extraordinary 7–10% of total BF mRNA, and we estimate that splicing rates at the active *VSG* are a staggering 512-fold higher than at the average Pol II-transcribed gene. Our data suggest that the proximity of these different nuclear bodies to the active ES facilitates the extraordinarily high localised demand for splicing components at this transcription unit.

Maintenance of high levels of splicing at the active ES is essential, and we discovered a feedback loop whereby splicing facilitates processive ES transcription. Inhibition of splicing with anti-U2 snRNA Morpholinos, sinefungin or SL-RNA depletion via SLAP1 RNAi, resulted in defective transcription elongation at the active ES. This indicates that ES transcription and splicing are coupled, whereby splicing enhances transcription elongation at the active ES. Why is ES transcription elongation perturbed when *trans*-splicing is blocked? Possibly, in the absence of splicing to process nascent transcripts, their accumulation at the ES transcription unit impedes efficient transcription elongation by Pol I.

Alternatively, there could be a more specific feedback mechanism. In other eukaryotes there is a direct coupling of Pol II transcription elongation with RNA splicing, with feedback between the two processes[35]. In yeast, a Pol I feedback loop is seen at the rDNA, where high levels of Pol I transcription elongation are functionally coupled with efficient pre-rRNA processing[36,37]. The mechanism underlying the possible feedback between *trans*-splicing and transcription elongation in *T. brucei* will require further investigation. Silent ESs are not in the proximity of the active ES[38], or associated with assemblies of splicing factor bodies and the low amounts of transcripts generated from silent ESs are not efficiently spliced or polyadenylated[39]. This suggests that silent ESs do not have access to sufficient pre-mRNA processing components, possibly provided by these nuclear bodies.

One important link between splicing and the active ES is the previously documented inter-chromosomal interaction between the active ES and the SL-RNA arrays[22]. Here, the active ES was shown to associate with a 'splicing compartment' containing at least one SL gene array, VEX1 and the SL-RNA gene transcription factor SNAP42. This 'splicing compartment' appears to

correspond to the SLAB described here, which contains the SL-RNA transcription factor SNAP3, VEX1 and the newly identified SLAP1, a novel regulator of SL-RNA production. The interaction of the active ES with the SLAB may allow direct access to the largest pool of SL RNA in the cell immediately after it is transcribed and properly matured.

However, SL RNA is not used in isolation in the cell to catalyse splicing. *Trans*-splicing of the SL RNA exon is catalysed by snRNAs packaged into snRNPs, creating splicing competent particles[40]. In addition to the SLAB, we identified two nuclear splicing factor bodies containing proteins with conserved functions in snRNP assembly in other eukaryotes. First, we identified an apparent canonical Cajal body (CB) in both BF and PF *T. brucei*. CBs are factories for RNP particles involved in splicing (snRNPs), ribosome biogenesis (snoRNPs) and telomere maintenance, with NOP56 and fibrillarin involved in snRNP/snoRNP maturation[10]. *T. brucei* normally has a single CB, unlike mammalian cells which have multiple CBs. This contained the conserved CB proteins NOP56 and fibrillarin, which are localised in both the CB and the nucleolus.

In addition, we identified a novel NUFIP body (NB). The relatively spherical nature of this body, in addition to its arrangement in an intriguing "shell" structure with internal compartments, could indicate that it is formed through phase separation[24]. Phase separation plays a key role in the assembly of multiple nuclear bodies including the nucleolus, which is a multiphase condensate[41,42]. The NUFIP body appears to be a putative splicing factor body containing the conserved proteins NUFIP and ZNHIT3. In mammalian cells, NUFIP in conjunction with splicing factors including ZNHIT3, facilitates assembly of snRNPs, as well as snoRNPs[25,26,43]. NUFIP (Rsa1 in yeast) appears to hold together the core components of the mature RNP, and links RNP assembly with Hsp90, as well as being required for snoRNA biogenesis[43]. Although snRNA modification and snRNP assembly typically occur in CBs, it is unclear if NUFIP in other organisms is localised in CBs or in a discrete NB[44–46]. In *T. brucei*, proteins corresponding to NUFIP, NufB1 and NufB2, have been shown to co-purify with the CRK9 kinase, which is essential for the first step of SL *trans*-splicing[32]. This all argues that the *T. brucei* NUFIP body could be involved in producing splicing components including snRNP assembly and/or SL RNA modification.

A CB was observed in only 26% of BF *T. brucei*. However, this body would be undercounted if the CB is either in very close proximity to or obscured by the nucleolus. Unfortunately, our CB markers also localised in the nucleolus, so we were not able to address this point. Alternatively, it is possible that the canonical function of the CB is split between the NUFIP, SLAB and Cajal bodies in *T. brucei*. This could facilitate optimal levels of SL RNA production and modification, as well as snRNP and snoRNP assembly. Cajal bodies have been shown to have shared components with other nuclear bodies, including the histone locus, Gem and promyelocytic (PML) bodies, as well as the nucleolus[10]. The *T. brucei* NB, SLAB and CB did not appear to share components, as investigated with the markers described here. However, nuclear body composition would need to be determined to further investigate this point. The relative proximity of the NB to the SLAB in both BF and PF *T. brucei* could suggest that the NB provides splicing components to the SLAB, thereby facilitating high rates of *trans*-splicing. The movement of splicing components between these nuclear bodies and the active ES could create an assembly line for the maturation of RNA processing components, thereby providing the active ES with the highest localised concentration of splicing factors in the cell.

CBs are dynamic structures, which in humans are present in cell types with high levels of transcriptional activity, including embryonic cells and transformed cancer cells[11]. CBs have been shown to preferentially localise near highly transcribed genes

including the histone genes, and the U snRNA and snoRNA loci, which form intra- and inter-chromosomal clusters[12]. Here, the CBs are thought to increase the levels and processivity of factors critical for efficient RNA splicing[11]. In BF *T. brucei*, the association of the CB and related nuclear splicing factor bodies with the ESB appears to be linked to the high demand for splicing at the active ES. In PF *T. brucei* the Pol I-transcribed procyclin locus (with a much lower demand for splicing factors than the active ES) was not obviously associated with the SLAB or NUFIP bodies.

These data all provide a new mechanistic explanation for monoallelic exclusion operating at the active ES in BF *T. brucei*. In most BF *T. brucei* cells in G1, only one NB or CB is present when these bodies are visible, and the majority of BF cells contain a single SLAB. These multiple nuclear splicing factor bodies appear to often cluster at the active ES, forming an ES nuclear body assembly facilitating the high levels of pre-mRNA splicing essential for efficient processive ES transcription. The large variation in distances of these bodies to the active ES could be a consequence of the dynamic nature of their interactions with it. As the number of each of these nuclear bodies is limiting (with frequently only one per cell), the ES 'winner takes it all' model[19] could incorporate winning access to the restricted number of nuclear bodies that we describe here. This could thereby ensure that sufficient pre-mRNA processing, and efficient Pol I transcription elongation, can only occur at a single active ES where these bodies all simultaneously congregate.

## Methods

**Culture and generation of *T. brucei* cell lines.** All experiments used the *T. brucei* 427 strain, with BF *T. brucei* cultured in modified HMI-9 medium (Thermo; 07490915 N) supplemented with 15% FCS (Gibco; 10270106) and 0.2 mM β-mercaptoethanol (Sigma; 63689) at 37 °C in 5% $CO_2$. The PF *T. brucei* were Amsterdam wild type or pSmOx cells[47] (gift from Jack Sunter), and were cultured in SDM-79 (Thermo; 07490916 N) with 10% FCS and 5 µg ml$^{-1}$ hemin (Sigma; H9039). Transgenic cell lines were generated by transfecting 10 µg of linearised DNA in Roditi Tb-BSF transfection buffer[48], using an Amaxa Nucleofector II with programme X-001 for BF or X-014 for PF *T. brucei*. Transformants were selected by the addition of the appropriate drug 6 h or 16 h after transfection for BF and PF cells, respectively. Cellular localisation of each protein studied was confirmed independently using at least two different epitope tags. Correct construct integration was confirmed using PCR. RNAi was induced with tetracycline (Sigma; T7660) at a final concentration of 1 µg ml$^{-1}$. Cell density was determined using a Neubauer haemocytometer. All cell lines used in this study are listed in Supplementary Table S4.

**DNA constructs.** Endogenous epitope tagging of proteins was achieved via homologous recombination using the pENT5 vector[49] (gift of Bill Wickstead). For N-terminal tagging, the first 414 bp of the coding sequence (CDS; after the ATG) and the first 307 bp of the 5'UTR were used for targeting. For C-terminal tagging, the last 414 bp of the CDS (excluding the stop codon) and the first 307 bp of the 3'UTR were used for targeting. All target fragments were amplified with Q5 DNA polymerase (NEB; M0491) from *T. brucei* 427 genomic DNA using standard conditions. The epitope tags were obtained from the following sources: 3xHA tag and 6xHA tag were synthesised (Thermo), TdTomato was from Addgene (plasmid no. 30530)[50], mCherry was amplified from the MK234 plasmid[8], Halo tag sequence was obtained from Addgene (plasmid no. 29644; gift from Eric Campeau) and mNeon green was amplified from the pPOTv4-mNG plasmid (gift of Sam Dean)[51]. A glycine-serine spacer of at least 15 nt was also inserted in between the epitope tag and the protein of interest. VEX1 epitope tagging was performed using pNAT-VEX1::12xmyc (gift of David Horn)[19].

Inducible transcript knockdown was performed using the pDEX-V4-Pex11 RNAi construct (gift of Bill Wickstead)[52]. Sense and anti-sense fragments of at least 500 bp of the target gene CDS were cloned flanking Pex11 using conventional cloning (Supplementary Table S5). The pDex-V4 vector integrates into a minichromosome containing the single-copy VSG-G4[53]. RNAi targets were selected using RNAit[54].

MS2 repeat containing plasmids were generated by inserting a 24× MS2 repeat sequence (Addgene plasmid no. 27120)[55] flanked by upstream and downstream target fragments corresponding to the locus of interest. Plasmids targeting the 221 ES promoter and telomere region and tubulin locus are described in refs. [8,14,17]. To target the region downstream of an endogenous rDNA promoter, rDNA target fragments were amplified from a plasmid with a single rDNA array.

**Inhibition of splicing and transcription**. In order to inhibit *trans*-splicing, cells were incubated various times with sinefungin (Millipore; 567051) to a final concentration of 2.5 µg ml$^{-1}$, and then processed for RNA-FISH or fluorescence microscopy. *Trans*-splicing was also inhibited using Morpholino oligonucleotides (Gene Tools) against the U2 snRNA (Tb927.2.5680) with the sequence: 5'-TGA-TAAGAACAGTTTAATAACTTGA-3'. As a control, pooled random 25-mer Morpholinos were used. Morpholinos were reconstituted in Roditi Tb-BSF transfection buffer[48]. For transfection with Morpholinos, $1 \times 10^7$ cells were centrifuged and resuspended in 100 µl Roditi Tb-BSF transfection buffer containing 80 µM Morpholinos. Cells were transferred to an electroporation cuvette and transfected with an Amaxa nucleofector II using programme X-001. Samples were then harvested for microscopy or RNA extraction as described.

In order to inhibit RNA Pol I transcription, cells were treated with 1 µM BMH-21 (Sigma; SML1183) dissolved in DMSO (Sigma; D8418). For BMH-21 washout, cells were washed twice in drug-free HMI-9 medium and resuspended in HMI-9 before direct fixation in media as described. For inhibition of transcription after splicing inhibition, cells were first treated with 2.5 µg ml$^{-1}$ sinefungin for 1 h and subsequently treated with 10 µg ml$^{-1}$ Actinomycin D (Sigma; A1410).

**qPCR**. mRNA expression levels were analysed using quantitative reverse transcriptase PCR (qRT-PCR). Total RNA was isolated from *T. brucei* cells under RNAse-free conditions using RNeasy kits (Qiagen; 74104), followed by DNase treatment with the TURBO DNA-free kit (Invitrogen; AM1907). Reverse transcription of cDNA was performed with the Omniscript RT kit (Qiagen; 205111) using 100 ng of template RNA and 5 mM random hexamers (Promega; c1181) according to the manufacturer's instructions. cDNA was diluted tenfold before use in qPCR reactions with Brilliant II SYBR Green qPCR Low ROX Master Mix (Agilent technologies; 600830). qPCR reactions were performed with a 7500 Fast Real-Time PCR system (Life Technologies).

For Morpholino transfection experiments, time points for genes of interest were normalised against the untreated sample and plotted as a relative change to the initial time point. Transcript quantitation after SLAP1 RNAi was performed after normalisation against actin for the respective time point. All primers used for qPCR are in Supplementary Table 2.

**RNA fluorescent in situ hybridisation (FISH) experiments**. For RNA-FISH, BF *T. brucei* cells were fixed directly in media for 10 min at room temperature (RT) using a 1:1.6:0.3 ratio of cells in suspension, 1× PSG and 37% (w/v) paraformaldehyde (PFA, Sigma; 47608). PF cells were fixed directly in media for 10 min at RT using a 1:8:1 ratio of cells in suspension, 1× PBS (Sigma; P4417) and 37% (w/v) PFA. Cells were next washed twice in 1× PBS before settling on to slides for 30 min. For combined RNA-FISH with immunofluorescence, slides were permeabilised with 0.1% NP-40 (Thermo; 28324) for 5 min at RT. Cells were then washed twice in 1× PBS, blocked in 1% BSA (Sigma; B4287), incubated with primary antibody for 1 h, washed in 1× PBS and incubated with secondary antibody. All antibodies were diluted in 1% BSA with 25 U of RNase inhibitor (Thermo; 10777019). The antibodies used were from Abcam: anti-mCherry (ab167453, 1:2000), anti-myc (ab9106, 1:1000) and anti-HA (ab18181, 1:750). The highly SUMOylated focus was visualised using an anti-SUMO antibody (kind gift of the Miguel Navarro lab)[34]. The anti-SUMO antibody was used undiluted, and blocking was performed with 5% normal goat serum (Sigma; NS02L) and 3% BSA with 25 U of RNase inhibitor. After incubation with secondary antibody, slides were washed in 1× PBS and then post-fixed in 4% (w/v) PFA for 10 min at RT. RNA-FISH was then completed as described in ref. [8]. Stellaris RNA-FISH probe sequences to target the VSGΨ and MS2 repeats are described in ref. [8], probes for the U2 snRNA and SL RNA intron are listed in Supplementary Table 6.

**Microscopy and image analysis**. Immunofluorescence was performed essentially as described above, with the exception that RNase inhibitor was not added to the BSA solution, cells were fixed in 2% PFA, and after incubation with secondary antibody, slides were washed and directly mounted in Vectashield with DAPI (Vector Laboratories; H-1200). The secondary antibodies used were either goat-anti-mouse AlexaFluor647 (Invitrogen; A21236), goat-anti-rabbit-AlexaFluor594 (Invitrogen; A11037), goat-anti-mouse AlexaFluor488 (Invitrogen; A11001) or goat-anti-mouse DyLight594 (Invitrogen; 35510). All secondary antibodies were used at a dilution of 1:500 in 1% BSA. The nucleolar marker L1C6 was visualised with an anti-L1C6 antibody[56] (1:100, gift from the Gull lab). The highly SUMOylated focus was visualised as described above using an anti-SUMO antibody. Widefield microscopy was performed with a Zeiss AxioImager M2 using a Hamamatsu ORCA-Flash4 camera with Zeiss Zen software. Anti-U2 snRNA Morpholino and sinefungin RNA-FISH experiments were imaged with a Zeiss AxioImager M1 using an AxioCam MRm camera with AxioImager software. All Z-stacks were taken in 0.2 µm increments.

Super-resolution structured illumination microscopy (SR-SIM) was performed using a Zeiss Elyra PS.1 microscope with a sCMOS PCO Edge (SIM) camera using Zeiss Zen software. Images were acquired with z-stacks taken in 0.1 µm increments with five phases and three grid rotations. SIM reconstruction was performed using Zen software after correcting for chromatic aberrations using the channel alignment function in Zen. 100 nm Tetraspeck beads (Thermo) were adhered to slides and were used to determine channel alignment for each experiment. SR-SIM

data quality was assessed for each experiment using SIM check[57]. Quantitation of distances between nuclear foci was performed essentially as described in ref. [8].

For measurements between RNA-FISH signals and proteins in which more than one protein puncta was present, the closest protein puncta measurement was used. The number of foci in cells was manually scored for all experiments. For quantitation experiments, images were acquired using the same settings and exposure times. For quantitation of nuclear body diameters, a maximum intensity projection was obtained from SR-SIM images, foci were manually delimited in ImageJ and the Feret's diameter (longest diameter in focus) was then calculated.

**HaloTag labelling**. Cells with endogenously tagged Halo::RPA2 were labelled with 200 nm of Janelia Fluor 646 HaloTag ligand (JF646; Promega; GA1120) by replacing one-fifth of the culture media with 1 µM JF646. Cells were then incubated for 30 min at 37 °C and harvested for microscopy after fixation in media. For combined antibody staining with Halo::RPA2 imaging, samples were permeabilised with 0.1% NP-40 for 2 min and processed as described above.

**Estimation of splicing rates**. mRNA production rates for the active VSG ES were estimated using available data for BF *T. brucei* Pol II-transcribed mRNA abundance and half-lives[13], as well as active VSG-ES mRNA abundance and half-lives[14,58,59]. ESAG mRNA half-lives were set to the median mRNA half-life in BF cells. Half-lives longer than 120 min were set at 120 min. mRNA copy numbers per cell were derived from mRNA abundance and using a previous estimate for 20,000 total mRNA molecules per BF cell[60] with VSG mRNA accounting for 10% of the total.

The number of mRNA molecules produced for 60 min necessary to maintain steady-state copy number levels (mRNA production rates) was calculated by first determining the number of mature mRNAs produced for one half-life, which is equal to half of the mRNA copy number per cell. To calculate the number of mRNA molecules produced for an hour, this value was then divided by the mRNA half-life (in min) and multiplied by 60. Finally, the number of mRNA molecules produced for 60 min was divided by the number of annotated alleles for an individual gene. All raw values used to calculate the mRNA production rates are listed in Supplementary Tables 1 and 2. The rates for productive SL *trans*-splicing events (resulting in the generation of mature mRNA) for a specific allele correspond to the mRNA production rates, because a single SL RNA molecule is consumed during the 5'-end processing of the transcript. The relative splicing efficiency of VSG pre-mRNA was calculated as the ratio between mRNA production rates for VSG and a typical *T. brucei* Pol II-transcribed gene.

**Bioinformatic and statistical analyses**. Potential *T. brucei* nuclear body proteins were initially identified using the TrypTag genome-wide protein localisation resource (http://tryptag.org)[18]. Homologues of relevant *T. brucei* proteins were determined using PSI-BLAST for up to five iterations of protein sequences from the TryTrypDB database. Relevant hits were used for additional PSI-BLAST searches and reciprocal BLAST searches. Phylogenetic trees were generated using MEGA X.[61]. Initial tree(s) for the heuristic search were obtained automatically by applying Neighbour-Join and BioNJ algorithms to a matrix of pairwise distances estimated using the JTT model, and then selecting the topology with superior log likelihood value. Multiple sequence alignments were performed using ClustalO under standard settings in Jalview.[62]. Relevant alignments were analysed with MEME to identify conserved sequence motifs[63], which were then analysed using PFAM.

All statistical analyses were performed using GraphPad Version 7.05. Violin plots were generated using R version 4. For transcription inhibition and splicing inhibition experiments and RNAi experiments, statistical significance was determined using paired two-tailed Student's *t* test, and considered significant when $P < 0.05$.

**Resources and reagents**. All unique materials generated in this study are available on request.

**Reporting summary**. Further information on research design is available in the Nature Research Reporting Summary linked to this article.

## Data availability
The datasets generated and analysed during the current study are available from the corresponding author on reasonable request. Source data are provided with this paper.

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

## Acknowledgements

The authors thank the TrypTag consortium, which is supported by the Wellcome Trust [108445/Z/15/Z]. for freely providing data that enabled part of this work. We thank Miguel Navarro (IPBLN-CSIC, Granada), Jay Bangs (University of Buffalo, USA), Piet Borst (Netherlands Cancer Institute), Keith Gull (University of Oxford, UK), Nina Papavasiliou (DKFZ, Heidelberg), Jack Sunter (Oxford Brookes University), Sam Dean (Warwick University), Bill Wickstead (University of Nottingham) and David Horn (University of Dundee, UK) for constructs or antibodies. We are grateful to Nancy Standart (University of Cambridge) for illuminating discussions. We thank Calvin Tiengwe, Agustina Berazategui and Dominic Marshall Sabey for discussions and comments on the manuscript. The Zeiss Elyra PS.1 super-resolution microscope used in this study was funded by the BBSRC grant BB/L015129/1 to the Facility for Imaging by Light Microscopy (FILM) at Imperial College London. GR is a Wellcome Senior Fellow in the Basic Biomedical Sciences. JB was funded by an Imperial President's PhD scholarship. This research was funded by the Wellcome Trust (095161) to GR and National Institutes of Health grants AI028798 and AI110325 to CT.

## Author contributions

J.B., N.K. and G.R. designed the experimental approach. J.B. and R.J. performed the experiments. J.B., N.K., C.T. and G.R. analysed the data. J.B. and G.R. wrote the manuscript.

## Competing interests

The authors declare no competing interests.
