## [Peer Review File · Nature Communications]

Reviewer comments, initial round - -

Reviewer #1 (Remarks to the Author):

Thoughts

The authors have identified a set of nuclear bodies associated with the ESB in BSF trypanosomes and highlight the importance of trans-splicing for generation of VSG transcripts. This is a comprehensive analysis of the positioning of these different nuclear bodies and is an important step to understanding how such high levels of VSG mRNA are achieved.

Points to be addressed:

1. The authors describe these bodies as dynamically associating with the active ES but what they mean by dynamic is not clear, the phrases "dynamic association" and "dynamically associated" are often used. What does this mean? No dynamics are shown. In most instances "associated" would probably be clearer and more accurate. One feels that this is a rather critical point. The authors are looking at a situation where the ESB and associated nuclear bodies are already formed and from their own previous work it appears that an inactive ES displaces the active ES at the ESB. Therefore, it appears that it is the ES not the nuclear bodies that are 'recruited' and dynamic. When the ESB is first formed, it is possible that these bodies are recruited to the activating ES but that is not addressed here.

2. Recent work including this have highlighted the complex nature of the nuclear region around the active ES and this is reflected in the changing terminology used to describe this region. In previous papers from the Horn group the SLAB and ESB are described as transcription compartments. Here the authors use the term conglomeration to describe the set of nuclear bodies around the active ES. Conglomerate seems like a poor fit as an analogy/descriptive term. The size of the nuclear bodies is also not explored. The key advantage of superresolution microscopy is that it reveals the size of the nuclear bodies, which are evidently significantly more widely spaced than their diameter. As conglomerate implies a co-embedding as part of a larger structure it doesn't fit well.

3. The authors show that the inhibition of splicing causes a reduction in transcription at the active ES. However, interestingly this effect was more pronounced for the ES telomere than promoter. Can the authors offer an explanation for this rather strange result? Does this mean that the initiation of transcription was able to continue to some degree and therefore is not/less reliant on splicing, likely representing an additional level of VSG expression control.

4. In a number of images of the SLAB proteins throughout the paper including 3A the signal appears to be elongated, with a dumbbell-like appearance. In the authors' analysis does this count as 1 or 2 foci? Does this shape suggest that both SL arrays are in close proximity in this nucleus?

5. What about SUMOylation? SUMOylation is central to PML nuclear body formation and implicated in aspects of Cajal body biology. The ESB sits in a highly SUMOylated focus - how does this relate to the various bodies described here?

6. Figure 9 shows a schematic of the arrangement of these different nuclear bodies but the authors should include the hyper-SUMOylated focus, which has been shown to be important by the Navarro group.

7. The shell pattern described for the localisation of various components of the NUFIP body should be elaborated on. From the images it appears that NufB1/2 form the outside of this structure with a core of NUFIP, RBP34 and ZNHIT34 - does that suggest a corraling function for NufB1/2? Was this pattern consistently observed? In addition, the phylogenetic analysis of NufB1/2 was interesting as neither tree followed the established pattern of kinetoplastid evolution, with the closest ortholog of NufB2 coming from *Angomonas* not *T. congolense* as might be expected. Are these proteins rapidly evolving?

8. The proportion of nuclei in which a Cajal body was much lower than for the other nuclear bodies. Does this suggest that the CB is not always required for ESB function or is this a technical problem with labelling or proximity to the nucleolus?

9. Here the authors show that VEX1 RNAi has minimal effect on SLAB and ESB positioning and this is consistent with the recent Faria et al., paper. In that work VEX2 was shown to be important for the positioning of the SLAB near the ESB and it would be informative to determine the role of VEX2 in the recruitment of the SLAB and the other bodies to the ESB; however, that is likely beyond the scope of this current work.

10. The images are not necessarily representative and, if so, the authors should consider representative images:

a. Many images of two bodies are paired with measurements of spacing, however the images typically show much smaller spacing than the mean spacing, see: Figs 2D vs E, 5C vs D and 6D vs E in particular. This somewhat misrepresents the proximity of the bodies relative to their size.
b. Images exclusively show nuclei with single bodies when two SLABs or NUFIPs are common, see: Figs 2C vs D and 4A vs C. Is there any tendency for the two bodies to be associated or spaced?

11. What is the expected spacing of two nuclear bodies if randomly positioned in the nucleus?

a. BF nuclei are typically 1.5 μm wide (radius around 750 nm) and nuclear bodies tend not to be right at the nucleus edge, excluded from the peripheral 200-300nm of the nucleus

b. Simulating the spacing of two points at purely random positions in a sphere of 500 nm radius gives a spacing of 400nm \pm 190nm

c. Simulating three bodies gives them lying within 500 nm of each other 40% of the time

I realise the procyclic form data in Fig 8 is a form of control, but procyclic form nuclei do tend to be larger.

Overall, I can accept that there is some degree of association for many of these bodies, but it should be tempered by the fact that these associations are only weakly higher than random.

Other points

12. Did the authors examine whether there was a loss of the ESB after the morpholino knockdown of U2?

13. For Figure 1c can the authors include signal intensity profile to highlight the U2 associated with the VSG transcript as this was not clear from the image.

14. Remove comment by CT in the figure legend of figure 3

15. Figure 7c – the label in figure says NUFIP but the legend says Cajal body

16. Page 2 para 2: Being an early-branching eukaryote does not imply unusual molecular biology, and much molecular biology has many shared features with all eukaryotes.

17. Page 3 para 2. Re-description of SL trans splicing is a little redundant.

18. Page 3 para 2. Mediated is an unusual word, given the mechanisms being investigated are largely about producing large quantities.

19. Page 3 para 3. This 'conglomerate' on average spans around 1/3 the diameter of the nucleus, calling it an assembly at the active VSG ES seems a step too far.

20. Page 4 para 1. A clear comparison with VEX1 would be beneficial, given that has intriguing associations with monoallelic expression and splicing.

21. Page 4 para 3. Use of the VSG pseudogene would benefit from a little more introduction/explanation as a clear unique sequence in BES1.

22. Page 5 para 1. It was unclear to me how qPCR was specific to nascent transcripts.

23. Page 6 para 3. Is there an existing terminology for the Pol II transcription factories assembled at the spliced leader foci? As prominent structures in many kinetoplastids I was a little surprised not to see an existing name.

24. Page 12 para 1. "Efficiently attract" is an unusual phrase - this implies a mechanism by which demand for splicing leads to a long-ranged attractive force.

Reviewer #2 (Remarks to the Author):

In this manuscript authors identified nuclear bodies associated with trans-splicing of VSGs. Using tryp-tag database, which shows the localization of proteins, they identified candidates that could be localized in nuclear bodies. Then, through tagging of these proteins, they sowed the presence of three nuclear bodies: Cajal Bodies, Spliced Leader Array Body (SLAB) and NUFIP body. These bodies are close to ESB (where VSG is expressed) and when VSG expression is blocked, the distance between ESB and these nuclear bodies increases. In addition, knock down of the

component of SLAB, SLAP1, blocks transcription of VSG.

Manuscript is well done, conclusions are supported by results and enriches the field of: nuclear organization and VSG expression.

There are, however, some points important to be addressed:

1. Manuscript is based on the calculation of mRNA production and trans splicing rate. However, the calculation of these data should be better explained. The rational of the formula bellow is not clear:

s [mRNA production rate] = [(mRNA copy number per cell) / 2] x [60 / (mRNA half-life in minutes) / number of alleles].

2. Authors use RNA-FISH and qPCR to measure nascent RNA. Although RNA-FISH is a methodology used to it, they should offer a short explanation why RNA-FISH detect nascent RNA and not stabilized RNA. Also, it is not clear in the manuscript if qPCR is detecting nascent or stabilized mRNA.

Figure 3B: It is not clear why the reduction of precursor mRNA of BES1 telomere is due to elongation reduction. Couldn't be due a higher rate of trans splicing?

Minor points:

The majority of double-expresser DE KW01 cells in G1 had a single NB (77 ± 7.5%) (Supplementary Fig. S8b). – Shouldn't be figure S7b?

The NB was not sensitive to inhibition of Pol I transcription using BMH-21 (Supplementary Fig. S9a, b), - I guess it is Figure S8.

Reviewer #3 (Remarks to the Author):

The subnuclear localization of VSG and its requirement for monoallelic expression has been a mystery since Chaves I et al. 1999 and Navarro M. et al 2001 first described the expression site body (ESB) in *T. brucei*. More recently, the Horn and Siegel laboratories have identified new molecules associated to VSG compartment (VEX1 and VEX2) and established spatial interaction between the ESB and the Spliced Leader locus. This work from the Rudenko lab identifies new factors that localize in close proximity to the ESB as new bodies: Cajal body, spliced leader array body (SLAB) and NUFIP body. Using super-resolution imaging, the authors make a careful characterization of the location of each body relative to the other in two stages of the parasite life cycle. The authors also use drugs, Morpholino and RNAi to obtain functional information about some of these factors and the relationship to splicing or ES-transcription.

Although I recognize the high value of the identification and validation of these proteins in the vicinity of the ESB, this paper does not provide solid functional results that allow us to better understand the link between trans-splicing and ESB. We still do not know why these bodies facilitate "the extraordinarily high demand for splicing at the active ES". As described below, I found that some conclusions about the role of these novel proteins in splicing to be speculative at this stage. It was also not clear how the role in elongation was established.

The manuscript could benefit from being re-organized so that the focus is more clear. Perhaps once all players are described in terms of their subnuclear localization and putative roles based on sequence identity, the authors could focus just on the most promising factor and perform a more detailed analysis, to provide more mechanistic information between splicing and BES processing.

MAJOR POINTS:

The authors say that the effect of anti-U2 Morpholinos is more pronounced close to the telomere than to the promoter. However the data in the graph suggests that the rate of decay is quite similar (Fig 1d). If both green and blue curves were normalized to 100, wouldn't the curves overlap? If they don't, is their difference statistically significant? This is an important point because I believe it is this result that allows authors to conclude that trans-splicing is required to maintain processive transcription elongation. Please elucidate.

The authors show that when SLAP1 is downregulated, the Spliced Leader RNA intron is no longer detected (Fig 3a). Couldn't this result be interpreted that SLAP1 acts as transcription factor

required for SL transcription. It is not clear to me how the authors conclude that SLAP1 is required for trans-splicing. Moreover, the fact that the SLAP1 RNAi originates a phenotype at 16h similar to what is obtained upon splicing inhibition with Morpholinos could be a coincidence. Finally, SLAP1 downregulation leads to a rapid arrest of cell growth as soon as 8h post-induction, which means that at 16h there could be already several indirect effects contributing to the phenotype described by the authors. Therefore the conclusion that "SLAP1 therefore appears to be a novel regulator of SL RNA production, necessary for the trans-splicing facilitating processive ES transcription." is an over-statement.

The authors clearly show that NUFIP is associated to the actively transcribed BES and other bodies. However, the conclusion that it may be involved in snRNP and/or SL RNA modification is just a prediction from sequence conservation. The authors do not show the function of this protein.

MINOR QUESTIONS:

Fig. 1 It is not clear how the rate of transcription and rate of splicing were calculated. A formula is given in the Materials and Methods, but the rationale should be explained in the results section.

Given that U2 is a general splicing factor, it is not intuitive to understand how its blockade with Morpholinos causes an effect only on BES loci. Is it because BES dependent quantitatively more on U2? This hypothesis should be put forward.

Sup Fig S2. Why do authors call these genes "nascent"? Did they use primers outside the transcript sequence? This should be explained in main text or figure legend.

Fig 1f. Not all bars reach 100%, suggesting that some cells do not have neither 1 ESB, nor diffuse PolI foci. What is their profile?

Fig. 9d. The effect of the drug is really subtle. Although statistically significant, is it biologically relevant?

ΨREVIEWER COMMENTS

We would like to thank the reviewers for reading our manuscript so carefully and coming with many good suggestions. We are glad to see that their response is so positive. We have now included new data (new Figure 8) and have modified other figures and the text as detailed below.

Reviewer #1 (Remarks to the Author):

Thoughts

The authors have identified a set of nuclear bodies associated with the ESB in BSF trypanosomes and highlight the importance of trans-splicing for generation of VSG transcripts. This is a comprehensive analysis of the positioning of these different nuclear bodies and is an important step to understanding how such high levels of VSG mRNA are achieved.

We were very gratified to see that Reviewer 1 finds that this a comprehensive analysis and an important step forward.

Points to be addressed:

1. The authors describe these bodies as dynamically associating with the active ES but what they mean by dynamic is not clear, the phrases "dynamic association" and "dynamically associated" are often used. What does this mean? No dynamics are shown. In most instances "associated" would probably be clearer and more accurate. One feels that this is a rather critical point. The authors are looking at a situation where the ESB and associated nuclear bodies are already formed and from their own previous work it appears that an inactive ES displaces the active ES at the ESB. Therefore, it appears that it is the ES not the nuclear bodies that are 'recruited' and dynamic. When the ESB is first formed, it is possible that these bodies are recruited to the activating ES but that is not addressed here.

Reviewer 1 is of course absolutely right that there is no data directly supporting a dynamic interaction of these nuclear bodies with the active ES. This phrasing reflects our own interpretation of the data. We have now replaced 'dynamically associated' or 'dynamic association' with 'associated' or 'association' as suggested.

2. Recent work including this have highlighted the complex nature of the nuclear region around the active ES and this is reflected in the changing terminology used to describe this region. In previous papers from the Horn group the SLAB and ESB are described as transcription compartments. Here the authors use the term conglomeration to describe the set of nuclear bodies around the active ES. Conglomerate seems like a poor fit as an analogy/descriptive term.

Reviewer 1 is correct that the word conglomerate implies that various bodies are glued together as seen in conglomerate stones. This is of course not the case. We have

now removed the word 'conglomerate' and refer instead to an 'assembly of nuclear bodies'.

The size of the nuclear bodies is also not explored. The key advantage of superresolution microscopy is that it reveals the size of the nuclear bodies, which are evidently significantly more widely spaced than their diameter. As conglomerate implies a co-embedding as part of a larger structure it doesn't fit well.

We have reanalysed our SR-SIM data and now include a violin plot with the calculated diameters of the NUFIP body, Cajal body, SLAB and ESB. This is now included in Supplementary Figure 10c. The corresponding figure legend has been modified, as well as the main text on p. 13. The materials and methods section has also been updated on p.23 to include details on how this analysis was done.

3. The authors show that the inhibition of splicing causes a reduction in transcription at the active ES. However, interestingly this effect was more pronounced for the ES telomere than promoter. Can the authors offer an explanation for this rather strange result? Does this mean that the initiation of transcription was able to continue to some degree and therefore is not/less reliant on splicing, likely representing an additional level of VSG expression control.

We do not think that inhibition of splicing is affecting initiation of Pol I transcription. Instead, we think that elongation of the extending RNA polymerase I could be partially impeded if nascent transcripts are not rapidly removed through trans-splicing and remain associated with the DNA template. If there is a partial Pol I elongation block, then fewer and fewer Pol I molecules would be present on the template the further that one proceeds down the transcription unit. One would therefore expect that the degree of reduction in transcript levels after a trans-splicing knockdown would become more pronounced the further a sequence is from the promoter. Transcripts from the ES telomere would therefore disappear more quickly than ES transcripts near the promoter.

Alternatively, there could be a direct coupling of transcription elongation with RNA splicing (and feedback between the two processes) as has been described for Pol II transcription in other eukaryotes (Saldi et al 2016 *J Mol Biol* 428, 2623). However, our phenomenon would need to be mechanistically completely different, as it involves Pol I rather than Pol II.

We have now introduced the following text on p. 5

Strikingly, this *trans-splicing* block resulted in a rapid reduction in cells with ES derived nascent transcripts. This was more pronounced for transcripts from the ES telomere compared with the ES promoter (Fig. 1d and Supplementary Fig. 1b). This argues that although Pol I transcription still initiates when *trans-splicing* is blocked, elongation is partially inhibited. As BES1 extends for more than 60 kb¹⁶, a transcript from the BES1 telomere could be expected to be affected more by a partial elongation block than one from the promoter region. This apparent block in transcription elongation is possibly caused by the accumulation of nascent unspliced transcript impacting on Pol I processivity.

and in the Discussion on p. 17

Why is ES transcription elongation perturbed when *trans*-splicing is blocked? Possibly, in the absence of splicing to process nascent transcripts, ~~its~~ their accumulation at the ES transcription unit impedes efficient transcription elongation by Pol I.

Alternatively, there could be a more specific feedback mechanism. In other eukaryotes there is a direct coupling of Pol II transcription elongation with RNA splicing, with feedback between the two processes³⁴. ~~Interestingly, a superficially similar~~ In yeast, a Pol I feedback loop is seen at the rDNA ~~in yeast~~, where high levels of Pol I transcription elongation ~~is~~ are functionally coupled with efficient pre-rRNA processing^{35,36}. The mechanism underlying the possible feedback between *trans*-splicing and transcription elongation in *T. brucei* will require further investigation.

4. In a number of images of the SLAB proteins throughout the paper including 3A the signal appears to be elongated, with a dumbbell-like appearance. In the authors' analysis does this count as 1 or 2 foci? Does this shape suggest that both SL arrays are in close proximity in this nucleus?

SLAB foci which appeared elongated were counted as a single, as two separate signals were not visible. Two foci were only counted if there were two clearly separate signals.

With regards to the proximity of the two SL arrays to each other, we have now included quantitation of the distance between the two SLAB foci when present within a single nucleus as Supplementary Figure S4B.

The corresponding figure legend has been updated and the text on p. 7 has been modified with the following text:

In G1 cells that contained two SLAB foci, these foci were usually separated with a mean distance of 870 ± 345 nm (Supplementary Fig. S4b).

5. What about SUMOylation? SUMOylation is central to PML nuclear body formation and implicated in aspects of Cajal body biology. The ESB sits in a highly SUMOylated focus - how does this relate to the various bodies described here?

This is a good point. SUMOylation has indeed been shown to be important in the formation of different nuclear bodies. We have now determined the localisation of the highly SUMOylated focus (HSF) in relation to the ESB, Cajal body, SLAB and NUFIP body. These data have now been put in a new Figure 8. From these data we find that the ESB is associated with the HSF, as shown previously in Lopez-Farfan *et al* 2014 *PLoS Pathogens*. In addition, using RNA-FISH experiments we find that the HSF is

in closer proximity to nascent RNA from the active ES promoter compared with the telomere. This result is in agreement with the CHIP results presented by Lopez-Farfan et al (2014) who showed a relative enrichment of SUMO at the active ES promoter compared with the telomere.

The following text on p. 13 and 14 has been added:

In BF *T. brucei* a highly SUMOylated focus (HSF) has previously been shown to be partially colocalised with the ESB³³. We determined the location of the HSF relative to the different BF *T. brucei* nuclear bodies (Fig. 8). We found that as previously documented, there is significant overlap between the ESB as visualised with Pol I (RPA2-Halo) and the HSF (Fig. 8a). The mean distance between the centre of these two foci was 110 ± 63 nm, which was slightly further than the centre of the VEX2 focus to the ESB (71 ± 54 nm) (Fig. 8 a, c). Interestingly, using combined immunofluorescence/ RNA-FISH experiments, we found that the HSF was in closer proximity to nascent RNA derived from the active ES promoter compared with the active ES telomere (113 ± 76 nm versus 248 ± 108 nm respectively) (Fig. 8b, c). In contrast, the centre of the ESB was almost equidistant to both active ES promoter and telomere transcripts (128 ± 88 nm and 151 ± 99 nm, respectively). The HSF did not appear to overlap with the Cajal body, SLAB or NUFIP body (Fig. 8d, e) with mean distances of 490 ± 301 nm, 317 ± 295 nm and 279 ± 251 nm respectively (Fig. 8f). These data collectively suggest that the HSF is tightly associated with the ESB, and is particularly close to the active ES promoter, possibly as the active ES promoter region is highly enriched for SUMOylated proteins.

In addition we have updated the Materials and Methods on p. 23:

The highly SUMOylated focus was visualised using an anti-SUMO antibody (kind gift of the Miguel Navarro lab)³³. The anti-SUMO antibody was used undiluted, and blocking was performed with 5% normal goat serum (Thermo) and 3% BSA with 25 U of RNase inhibitor.

As well as on p. 21

Halo tag sequence was obtained from Addgene (gift from Eric Campeau; plasmid no. 29644) and on p. 24-25

HaloTag Labelling

Cells with endogenously tagged Halo::RPA2 were labelled with 200 nM of Janelia Fluor 646 HaloTag ligand (JF646; Promega) by replacing one fifth of the culture media with 1 μ M JF646. Cells were then incubated for 30 mins at 37°C and harvested for microscopy after fixation in media. For combined antibody staining with Halo::RPA2 imaging, samples were permeabilised with 0.1% NP-40 for 2 mins. and processed as described above.

6. Figure 9 shows a schematic of the arrangement of these different nuclear bodies but the authors should include the hyper-SUMOylated focus, which has been shown to be important by the Navarro group.

We have now included the HSF in Figure 10. The corresponding figure legend has also been updated accordingly.

7. The shell pattern described for the localisation of various components of the NUFIP body should be elaborated on. From the images it appears that NufB1/2 form the outside of this structure with a core of NUFIP, RBP34 and ZNHIT34 – does that suggest a corraling function for NufB1/2? Was this pattern consistently observed?

This shell pattern within the NUFIP body was consistently observed. However, due to the very small size of this body (100-200 nm), the exact ultrastructure would need to be further investigated using an additional higher resolution microscopy technique such as STED.

We think that it is most likely that this shell-like structure is a consequence of phase separation resulting in components segregating into different compartments. This process plays an important role in the assembly of a number of nuclear bodies including the nucleolus. However, this would need to be experimentally tested.

We have modified the text on p. 10 to read:

Interestingly, higher magnifications indicated that the NB was further sub-compartmentalised into a “shell”-type structure (insets Fig. 4b-d). The relatively spherical structure of the NB, which is subdivided into subdomains, could indicate that it is formed through phase separation²³.

In addition, we have added the following text to the Discussion on p. 18:

Additionally, we identified a novel NUFIP body (NB). The relatively spherical nature of this body, in addition to its arrangement in an intriguing “shell” structure with internal compartments, could indicate that it is formed through phase separation²³. Phase separation plays a key role in the assembly of multiple nuclear bodies including the nucleolus, which is a multiphase condensate^{40,41}.

In addition, the phylogenetic analysis of NufB1/2 was interesting as neither tree followed the established pattern of kinetoplastid evolution, with the closest ortholog of NufB2 coming from *Angomonas* not *T. congolense* as might be expected. Are these proteins rapidly evolving?

We have double checked the phylogenetic analysis for NufB2 and have run additional types of phylogenetic analyses. These have found the same result that the *T. brucei* NufB2 is more similar to the *A. deanei* NufB2 than the *T. congolense* ortholog. It could be possible that this protein is rapidly evolving.

8. The proportion of nuclei in which a Cajal body was much lower than for the other nuclear bodies. Does this suggest that the CB is not always required for ESB function or is this a technical problem with labelling or proximity to the nucleolus?

We find similar numbers of CB when following this body with different proteins (such as FIB2 or NHP2). We also observe similar numbers of CB when using different epitope tags for the same protein. It is therefore not very likely that we are observing an artefact arising from a specific protein/ epitope tag.

However, false negatives would be obtained in our quantitation of the numbers of CB if the CB is in very close proximity to (or inside) the nucleolus. Unfortunately, we do not have a marker which is specific for just the CB so were not able to address this point.

If not all cells have a CB, another possibility could be that the function of a canonical CB in *T. brucei* can be shared with the NUFIP body and the SLAB. This could mean that the SLAB and NUFIP bodies are more important for the function of the ESB, with the CB playing a lesser role.

As all of the CB proteins we have identified are present in both the nucleolus and the CB, we can not precisely determine the effect of knockdown of the CB on the ESB, as we will also disrupt nucleolar function.

We have addressed these points by inserting the following text on p. 18:

A CB was observed in only 26% of BF *T. brucei*. However, this body would be undercounted if the CB is either in very close proximity to or obscured by the nucleolus. Unfortunately, our CB markers were also localised in the nucleolus, so we were not able to address this point.

9. Here the authors show that VEX1 RNAi has minimal effect on SLAB and ESB positioning and this is consistent with the recent Faria et al., paper. In that work VEX2 was shown to be important for the positioning of the SLAB near the ESB and it would be informative to determine the role of VEX2 in the recruitment of the SLAB and the other bodies to the ESB; however, that is likely beyond the scope of this current work.

We agree that this is an interesting question, however we feel that these experiments would be beyond the scope of this work and would require a separate investigation.

10. The images are not necessarily representative and, if so, the authors should consider representative images:

a. Many images of two bodies are paired with measurements of spacing, however the images typically show much smaller spacing than the mean spacing, see: Figs 2D vs E, 5C vs D and 6D vs E in particular. This somewhat misrepresents the proximity of the bodies relative to their size.

We have now replaced the microscopy images in Figures 2D, 5C and 6D with images which show cells where the mean spacing of the respective nuclear bodies is shown.

b. Images exclusively show nuclei with single bodies when two SLABs or NUFIPs are common, see: Figs 2C vs D and 4A vs C. Is there any tendency for the two bodies to be associated or spaced?

We have now included quantitation of the distance between the SLAB and NUFIP body in nuclei that contain two of the same body. For the SLAB, this is shown in supplementary figure S4B and for the NUFIP body this is shown in Supplementary figure S8A.

We have also included the following text on p. 7

In G1 cells that contained two SLAB foci, these foci were usually separated with a mean distance of 870 ± 345 nm (Supplementary Fig. S4b).

And on p. 10

In G1 cells that contained two NUFIP bodies, these foci were normally separated with a mean distance of 885 ± 358 nm (Supplementary Fig. S8a).

11. What is the expected spacing of two nuclear bodies if randomly positioned in the nucleus?

We have previously quantitated the distance between two unrelated nuclear compartments in the BF nucleus using RNA-FISH. Specifically, we measured the distance of nascent RNA derived from the tubulin locus (using MS2 repeats inserted into a single tubulin locus) to nascent RNA derived from the active 221ES telomere (using VSG Ψ). In this experiment, we found that the mean spacing between these two compartments was 0.88 ± 0.48 μ m and they were located within 500 nm of each other 17% of the time. The data for this experiment is shown in Budzak *et al.*, 2019 PNAS Figure 5B/ C. In contrast we observe the SLAB, ESB and NUFIP body lying within 500 nm of each other 59% of the time.

a. BF nuclei are typically 1.5 μ m wide (radius around 750 nm) and nuclear bodies tend not to

be right at the nucleus edge, excluded from the peripheral 200-300nm of the nucleus

b. Simulating the spacing of two points at purely random positions in a sphere of 500 nm radius gives a spacing of 400nm +/- 190nm

c. Simulating three bodies gives them lying within 500 nm of each other 40% of the time

I realise the procyclic form data in Fig 8 is a form of control, but procyclic form nuclei do tend to be larger.

Overall, I can accept that there is some degree of association for many of these bodies, but it should be tempered by the fact that these associations are only weakly higher than random.

We have now amended the main text to include the information discussed above to act as a reference point for the spacing of two randomly positioned compartments in the nucleus.

We have added the following text on p. 13:

We do not think that the frequent localisation of the CB, ESB, NB and SLAB within 500 nm of each other is fortuitous. The CB, ESB, NB and SLAB have relatively similar average diameters of around 235 nm, 287 nm, 222 nm or 317 nm respectively (Supplementary Fig. S10c). We have previously determined the proximity of the BES1 and the unrelated tubulin locus in BF *T. brucei*⁸, and showed that these two loci were separated from each other with an average distance of 880 ± 480 nm, and located within 500 nm of each other only 17% of the time. These data support the hypothesis that these nuclear bodies preferentially associate with the active ES.

Other points

12. Did the authors examine whether there was a loss of the ESB after the morpholino knockdown of U2?

We did not determine whether there was a loss of the ESB after transfecting cells with anti-U2 snRNA Morpholinos. However, we would expect this to produce the same result as inhibiting splicing through sinefungin treatment or SLAP1 RNAi. This data is already included in Figures 1e and 1f and 3e and 3f.

13. For Figure 1c can the authors include signal intensity profile to highlight the U2 associated with the VSG transcript as this was not clear from the image.

We have now modified Figure 1c to include a signal intensity profile of the fluorescence from the U2 snRNA and VSGΨ FISH.

14. Remove comment by CT in the figure legend of figure 3

This error has now been corrected, and the comment removed.

15. Figure 7c – the label in figure says NUFIP but the legend says Cajal body

The figure legend has now been corrected.

16. Page 2 para 2: Being an early-branching eukaryote does not imply unusual molecular biology, and much molecular biology has many shared features with all eukaryotes.

The text on p. 2 now reads:

Compared with more widely studied eukaryotes, *T. brucei* has highly unusual molecular biology features, including limited transcriptional control.

17. Page 3 para 2. Re-description of SL trans splicing is a little redundant.

We have changed the text on p. 3 to read:

These extraordinarily high expression levels are facilitated by high rates of transcription initiation by Pol I, whereby subsequent *trans*-splicing ~~of a capped SL RNA sequence~~ results in the generation of ~~functional a capped and translatable~~ mRNA⁹.

18. Page 3 para 2. Mediated is an unusual word, given the mechanisms being investigated are largely about producing large quantities.

We have now changed the text on p. 3 to read:

We investigated how these phenomenal levels of mRNA expression from a single *VSG* gene are ~~achieved mediated~~,

19. Page 3 para 3. This 'conglomerate' on average spans around 1/3 the diameter of the nucleus, calling it an assembly at the active VSG ES seems a step too far.

We have now rephrased this throughout the entire manuscript. We do not use the word 'conglomerate' and say that these nuclear bodies 'associate' with the active VSG ES.

20. Page 4 para 1. A clear comparison with VEX1 would be beneficial, given that has intriguing associations with monoallelic expression and splicing.

We weren't sure exactly which text the reviewer is referring to here.

21. Page 4 para 3. Use of the VSG pseudogene would benefit from a little more introduction/explanation as a clear unique sequence in BES1.

We have added the following text on p. 5:

The VSG Ψ is unique to BES1, and produces an unstable transcript with a half-life of 1.63 ± 0.65 min, which does not appear to leave the nucleus⁸.

22. Page 5 para 1. It was unclear to me how qPCR was specific to nascent transcripts.

We have now changed the text on p. 5 and 6 to read:

In addition to using RNA-FISH, unprocessed or nascent transcripts from these loci were also quantitated with qPCR, using one primer annealing outside of the transcript splice/processing site and the other annealing within the transcript itself.

23. Page 6 para 3. Is there an existing terminology for the Pol II transcription factories assembled at the spliced leader foci? As prominent structures in many kinetoplastids I was a little surprised not to see an existing name.

We were unable to find any specific terminology for the Pol II transcription factories assembled at the SL array loci.

24. Page 12 para 1. "Efficiently attract" is an unusual phrase - this implies a mechanism by which demand for splicing leads to a long-ranged attractive force.

We have changed the text on p. 15 to read:

Possibly these lower rates are not high enough to ~~efficiently attract~~ require nuclear splicing factor bodies to be in the immediate vicinity of the procyclin genes.

Reviewer #2 (Remarks to the Author):

In this manuscript authors identified nuclear bodies associated with trans-splicing of VSGs. Using tryp-tag database, which shows the localization of proteins, they identified candidates that could be localized in nuclear bodies. Then, through tagging of these proteins, they sowed the presence of three nuclear bodies: Cajal Bodies, Spliced Leader Array Body (SLAB) and NUFIP body. These bodies are close to ESB (where VSG is expressed) and when VSG expression is blocked, the distance between ESB and these nuclear bodies increases. In addition, knock down of the component of SLAB, SLAP1, blocks transcription of VSG. Manuscript is well done, conclusions are supported by results and enriches the field of: nuclear organization and VSG expression.

We were very happy to see that Reviewer 2 found that our manuscript was well done, with well supported conclusions which enrich the field.

There are, however, some points important to be addressed:

1. Manuscript is based on the calculation of mRNA production and trans splicing rate. However, the calculation of these data should be better explained. The rational of the formula bellow is not clear:

s [mRNA production rate] = [(mRNA copy number per cell) / 2] x [60 / (mRNA half-life in minutes) / number of alleles].

We have tried to make the rationale behind the calculations more clear. We have now removed the formula, and replaced it with a more extensive explanation of how these calculations were performed. This allows the reader to follow the train of thought and provides more clarity.

We have modified the text in the Materials and Methods on p. 25 to read:

The number of mRNA molecules produced for 60 min necessary to maintain steady-state copy number levels (mRNA production rates) were calculated by first determining the number of mature mRNAs produced for one half-life, which is equal to half of the mRNA copy number per cell. To calculate the number of mRNA molecules produced for one hour, this value was then divided by the mRNA half-life (in min) and multiplied by 60. Finally, the number of mRNA molecules produced for 60 min was divided by the number of annotated alleles for an individual gene: ~~as $[mRNA\ production\ rate] = [(mRNA\ copy\ number\ per\ cell) / 2] \times [60 / (mRNA\ half\ life\ in\ minutes) / number\ of\ alleles]$.~~ All raw values used to calculate the mRNA production rates are listed in Supplementary tables 1 and 2. The rates for productive SL *trans*-splicing events (resulting in the generation of mature mRNA) for a specific allele correspond to the mRNA production rates, because a single SL RNA molecule is consumed during the 5'-end processing of the transcript. The relative splicing efficiency of VSG pre-mRNA was calculated as the ratio between mRNA production rates for VSG and a typical *T. brucei* Pol II transcribed gene.

In addition we have inserted the following text on p. 4

We estimated the number of mRNA molecules produced per hour from different genes, using available data on mRNA abundance and half-lives^{13, 14}, (Fig. 1a, Supplementary Table 1 and 2, see Materials and Methods for explanation of calculation).

2. Authors use RNA-FISH and qPCR to measure nascent RNA. Although RNA-FISH is a methodology used to it, they should offer a short explanation why RNA-FISH detect nascent RNA and not stabilized RNA.

The MS2 repeats do not contain RNA processing signals, and the transcripts derived from them are not processed into mature mRNA but rapidly degraded. The VSG Ψ in BES1 also generates a highly unstable transcript. We have previously determined the half-lives of the RNA-FISH signal of transcripts transcribed from the MS2 repeats or the VSG Ψ within BES1. These are 1.75 ± 0.62 min and 1.63 ± 0.65 min, respectively. See Budzak *et al.*, 2019 PNAS figure 6B. Furthermore, RNA-FISH experiments show only a single nuclear focus is present when detecting either transcript, highlighting that there are no additional nuclear or cytoplasmic foci which would be expected for RNA-FISH experiments detecting mature mRNAs.

We have now inserted the following text on p. 5:

This U2-positive focus colocalised with nascent RNA at the active ES, detected with a VSG pseudogene (VSG Ψ) probe (Fig. 1c). The VSG Ψ is unique to BES1, and produces an unstable transcript with a half-life of 1.63 ± 0.65 min, which does not appear to leave the nucleus⁸.

and:

We monitored nascent transcripts from *T. brucei* loci where 24x MS2 repeat containing constructs were integrated. The 24x MS2 repeats do not contain RNA processing signals, and the transcripts derived from them are not processed into mature mRNA, and are rapidly degraded. We have previously shown that MS2 repeat containing transcripts generated from the active ES are highly unstable, with a half-life of 1.75 ± 0.62 min⁸. These 24x MS2 repeat marked loci included the promoter and telomeric regions

Also, it is not clear in the manuscript if qPCR is detecting nascent or stabilized mRNA.

In order to clarify this, we have now inserted the following text on p. 5.

In addition to using RNA-FISH, unprocessed or nascent transcripts from these loci were also quantitated with qPCR, using one primer annealing outside of the transcript splice/processing site and the other annealing within the transcript itself.

Figure 3B: It is not clear why the reduction of precursor mRNA of BES1 telomere is due to elongation reduction. Couldn't be due a higher rate of trans splicing?

In these RNA-FISH experiments, as we are detecting nascent RNA which is not trans-spliced, the rate of trans-splicing should not affect signal detection. Therefore, the main factor which would affect signal detection would be transcription elongation.

Minor points:

The majority of double-expresser DE KW01 cells in G1 had a single NB ($77 \pm 7.5\%$) (Supplementary Fig. S8b). – Shouldn't be figure S7b?

Supplementary figure 7b shows data for the NufB1 phylogenetic tree. Supplementary figure 8 includes data for the percentage of DE KW01 cells with a NUFIP body and this sentence therefore references the correct figure.

The NB was not sensitive to inhibition of Pol I transcription using BMH-21 (Supplementary Fig. S9a, b), - I guess it is Figure S8.

Supplementary figure S8 shows data for the percentage of DE KW01 cells with a NUFIP body. The effect of BMH-21 incubation on the NUFIP body is shown in Supplementary Figure S9 and this sentence therefore references the correct figure.

Reviewer #3 (Remarks to the Author):

The subnuclear localization of VSG and its requirement for monoallelic expression has been a mystery since Chaves I et al. 1999 and Navarro M. et al 2001 first described the expression site body (ESB) in *T. brucei*. More recently, the Horn and Siegel laboratories have identified new molecules associated to VSG compartment (VEX1 and VEX2) and established spatial interaction between the ESB and the Spliced Leader locus. This work from the Rudenko lab identifies new factors that localize in close proximity to the ESB as new bodies: Cajal body, spliced leader array body (SLAB) and NUFIP body. Using super-resolution imaging, the authors make a careful characterization of the location of each body relative to the other in two stages of the parasite life cycle. The authors also use drugs, Morpholino and RNAi to obtain functional information about some of these factors and the relationship to splicing or ES-transcription.

Although I recognize the high value of the identification and validation of these proteins in the vicinity of the ESB, this paper does not provide solid functional results that allow us to better understand the link between trans-splicing and ESB. We still do not know why these bodies facilitate “the extraordinarily high demand for splicing at the active ES”. As described below, I found that some conclusions about the role of these novel proteins in splicing to be speculative at this stage. It was also not clear how the role in elongation was established.

The manuscript could benefit from being re-organized so that the focus is more clear. Perhaps once all players are described in terms of their subnuclear localization and putative roles based on sequence identity, the authors could focus just on the most promising factor and perform a more detailed analysis, to provide more mechanistic information between splicing and BES processing.

We are very happy to see that Reviewer 3 recognises the high value of our identification and validation of novel nuclear bodies in *T. brucei* which are found in the vicinity of the ESB. We think that the extensive conservation of many of the newly identified proteins with those found in other species provides a likely prediction as to what these bodies do. A more detailed experimental analysis of the functional role of these bodies will need to be the basis of a future investigation.

We have modified the text of our manuscript, which we hope makes the focus more clear in this current version.

MAJOR POINTS:

The authors say that the effect of anti-U2 Morpholinos is more pronounced close to the telomere than to the promoter. However the data in the graph suggests that the rate of decay is quite similar (Fig 1d). If both green and blue curves were normalized to 100, wouldn't the curves overlap? If they don't, is their difference statistically significant? This is an important

point because I believe it is this result that allows authors to conclude that trans-splicing is required to maintain processive transcription elongation. Please elucidate.

We have repeated the analysis of the 221ES-promoter and telomere RNA-FISH signals after anti-U2 snRNA Morpholino transfection with data points normalised to 100. Although the slope of the curves are similar up to 15 minutes, they are significantly different after this time point and do not overlap. Due to the unconventional nature of this analysis for this type of experiment, we have included this figure for the rebuttal but have not updated the main figure to include normalisation to 100%. $p < 0.05 = *$, $p < 0.005 = **$.

The authors show that when SLAP1 is downregulated, the Spliced Leader RNA intron is no longer detected (Fig 3a). Couldn't this result be interpreted that SLAP1 acts as transcription factor required for SL transcription.

Yes this reviewer is correct that SLAP1 could indeed be a transcription factor.

The text on p. 8 has been modified to read:

The SL RNA intron was undetectable after induction of SLAP1 RNAi for 16 hours, suggesting that SLAP1 is essential for SL RNA transcription and/or production (Fig. 3a).

It is not clear to me how the authors conclude that SLAP1 is required for trans-splicing.

SLAP1 is necessary for production of the SL-RNA, which is in turn required for trans-splicing.

The text on p. 9 now reads:

SLAP1 therefore appears to be a novel regulator of SL RNA production. This continued SL RNA synthesis is necessary for the *trans*-splicing which facilitates ~~facilitating~~ processive ES transcription.

Moreover, the fact that the SLAP1 RNAi originates a phenotype at 16h similar to what is obtained upon splicing inhibition with Morpholinos could be a coincidence.

SLAP1 RNAi is rapidly lethal. It results in depletion of SL-RNA and an increase in unspliced pre-mRNA which is indicative that splicing has been inhibited.

Anti-U2 snRNA Morpholino transfection has been previously been shown to rapidly inhibit splicing in *T. brucei* resulting in an increase in unspliced pre-mRNA.

Due to the fact that both treatments result in splicing inhibition, we would argue that this phenotype does not represent a coincidence and is specific to splicing inhibition.

Finally, SLAP1 downregulation leads to a rapid arrest of cell growth as soon as 8h post-induction, which means that at 16h there could be already several indirect effects contributing to the phenotype described by the authors.

It is of course correct that blocking generation of the SL RNA through KD of SLAP1 would indeed lead to indirect effects.

Therefore the conclusion that “SLAP1 therefore appears to be a novel regulator of SL RNA production, necessary for the *trans*- splicing facilitating processive ES transcription.” is an over-statement.

The text on p. 9 now reads:

SLAP1 therefore appears to be a novel regulator of SL RNA production. This continued SL RNA synthesis is necessary for the *trans*-splicing which facilitates ~~facilitating~~ processive ES transcription.

The authors clearly show that NUFIP is associated to the actively transcribed BES and other bodies. However, the conclusion that it may be involved in snRNP and/or SL RNA modification is just a prediction from sequence conservation. The authors do not show the function of this protein.

The text has now been reworded to highlight that these functions are putative.

We have added the following text on p. 11:

These data argue that the possible function of the NB is that of a splicing factor body involved in snRNP assembly and/ or SL RNA modification.

and

In summary, our data describe a novel NUFIP body, which ~~has a possible function appears to be involved~~ in snRNP assembly and/ or SL RNA modification,

And on p. 18

The NUFIP body appears to be a putative splicing factor body containing the conserved proteins NUFIP and ZNHIT3.

on p. 17

In addition to the SLAB, we identified two nuclear splicing factor bodies containing proteins with conserved functions in snRNP assembly in other eukaryotes.

on p. 18

This all argues likely that the *T. brucei* NUFIP body is could be involved in producing splicing components

MINOR QUESTIONS:

Fig. 1 It is not clear how the rate of transcription and rate of splicing were calculated. A formula is given in the Materials and Methods, but the rationale should be explained in the results section.

This is a good point which was also raised by Reviewer 2. We have now removed the formula and replaced it with more extensive text on p. 25 in the Materials and Methods explaining how the calculations were performed.

Given that U2 is a general splicing factor, it is not intuitive to understand how its blockade with Morpholinos causes an effect only on BES loci. Is it because BES dependent quantitatively more on U2? This hypothesis should be put forward.

Virtually all of the Pol II transcribed loci depend on U2 snRNA for trans-splicing. However, the active BES is particularly dependent on it, as so much mRNA is generated from this transcription unit. In addition BF *T. brucei* is exquisitely sensitive to knock-down of VSG transcript, which results in a very rapid abrupt arrest.

We have now inserted the following text on p. 6

These data indicate that although virtually all Pol II transcribed loci in *T. brucei* require U2 snRNA for *trans*-splicing, the active BES is particularly dependent on it. This is presumably due to the much higher requirement for splicing (and therefore U2 snRNA) at the active BES compared with the average Pol II transcribed locus (Fig. 1b). In addition, BF *T. brucei* is exquisitely sensitive to knockdown of VSG transcript, which results in a very rapid growth arrest¹⁷.

Sup Fig S2. Why do authors call these genes “nascent”? Did they use primers outside the transcript sequence? This should be explained in main text or figure legend.

Additional information has been added to the main text to clarify that the primer binding sites include a primer which is outside of the sequence comprising the transcript mature mRNA

We have inserted the following text on p. 5 and 6:

In addition to using RNA-FISH, unprocessed or nascent transcripts from these loci were also quantitated with qPCR, using one primer annealing outside of the transcript splice/processing site and the other annealing within the transcript itself.

Fig 1f. Not all bars reach 100%, suggesting that some cells do not have neither 1 ESB, nor diffuse PolII foci. What is their profile?

In the remaining cells there is indeed no ESB.

We have added the following text to the figure legend of Figure 1f)

Values do not add up to 100% as not all cells had detectable Pol I signal.

Fig. 9d. The effect of the drug is really subtle. Although statistically significant, is it biologically relevant?

We have added the following text on p. 14 to clarify that this experiment requires further investigation.

Further experiments will be needed to fully determine how these nuclear bodies are recruited to the active ES.

As an additional point which was not raised by these reviewers, we have now designated the fibrillarin analysed in this manuscript as fibrillarin-2.

We have modified the following text on p. 12:

T. brucei appears to have three fibrillarin proteins. The fibrillarin we designated as fibrillarin-2 (Tb927.10.14750), as well as the conserved CB proteins NOP56 (Tb927.8.3750) were present in both nucleoli and CBs, presumably as they have a dual function in both rRNA and snRNA modification ³².

Reviewer comments, further round - -

Reviewer #1 (Remarks to the Author):

The authors have addressed all the points we had either with textual changes/tweaked figures or some additional analyses and have done the same for the other reviewers I'd say. I'm happy with the phrase assembly of nuclear bodies better than conglomerate.

Specifically, in terms of the positioning question. It's interesting that the standard deviations for many of the measurements are really quite large. These additional splicing related bodies can either be really quite close to each other/ESB in some cells or quite far away in others - might this indicate transient interactions or cell differences?

Overall, I think this is a good contribution to an increasingly active field.

Reviewer #2 (Remarks to the Author):

All questions were answered properly

Reviewer #3 (Remarks to the Author):

I thank the authors for addressing most of my concerns.

Reviewer comments

We would like to thank the reviewers once again for reading our manuscript so carefully and coming with many good suggestions. We feel that this has helped improve the manuscript from the first submission. We are glad to see that in our resubmission, we have been able to answer all questions raised and address concerns raised by reviewers.

Reviewer #1 (Remarks to the Author):

The authors have addressed all the points we had either with textual changes/tweaked figures or some additional analyses and have done the same for the other reviewers I'd say. I'm happy with the phrase assembly of nuclear bodies better than conglomerate.

Specifically, in terms of the positioning question. It's interesting that the standard deviations for many of the measurements are really quite large. These additional splicing related bodies can either be really quite close to each other/ESB in some cells or quite far away in others - might this indicate transient interactions or cell differences?

This is a very interesting point and one which will require further experiments. Based on the data we have, our interpretation is that these interactions are dynamic which results in differential positioning of these nuclear compartments relative to each other and the active ES. However, due to the difficulty of doing live cell imaging in BSF *T. brucei* (particularly for long periods of time) we cannot currently make a very strong statement on how dynamic/transient these interactions are. In addition, the exact mechanism of how these nuclear bodies are targeted to specific genomic loci is still unclear. As such, there may be heterogeneity between cells which regulates the frequency of interaction of these nuclear bodies with the active ES.

We have responded to this comment by inserting the following text on p. 19 and 20

These multiple nuclear splicing factor bodies appear to often cluster at the active ES, forming an ES nuclear body assembly facilitating the high levels of pre-mRNA splicing essential for efficient processive ES transcription. **The large variation in distances of these bodies to the active ES could be a consequence of the dynamic nature of their interactions with it.**

Overall, I think this is a good contribution to an increasingly active field.

We thank this reviewer for the positive comments on our manuscript.

Reviewer #2 (Remarks to the Author):

All questions were answered properly

We are pleased that we were able to answer all questions raised by this reviewer.

Reviewer #3 (Remarks to the Author):

I thank the authors for addressing most of my concerns.

We are pleased that we were able to address concerns raised by this reviewer.